# Robust Human-AI Complementarity under Uncertainty

**Yewon Byun** [1]   **Bryan Wilder** [1]

## Abstract

Machine learning models are often intended to augment rather than replace human decision-makers, by providing information that is complementary to human judgement. Yet, in practice, human decision makers routinely fail to realize such complementary gains, even when models provide useful signal. In this work, we study how asymmetric information about the quality of information available to a human decision maker vs. an AI impacts the ability of a decision maker to extract complementary value from AI predictions. We show that a key factor is the error correlation structure between human and AI predictions. In particular, when the AI's prediction errors are *negatively correlated* with those of the human, the decision-maker can construct robust strategies which guarantee improvements in expected utility. We empirically investigate whether these conditions for complementarity arise in practice, using real-world forecasting benchmarks.

## 1. Introduction

Many hope that machine learning models will augment human decision making without replacing it by providing decision makers with complementary information. However, attempting to rely appropriately on machine learning outputs can be difficult; human decision makers routinely fail to benefit from complementary signal in AI predictions (Vaccaro et al., 2024) even when such signal is available (Guo et al., 2024). Previous work has hypothesized that people may struggle to update their beliefs appropriately in light of AI predictions (Agarwal et al., 2023) or identify what new information the AI model provides (Guo et al., 2025). Here, we document another kind of limitation that makes it difficult for even a rational decision maker to benefit from complementary information: uncertainty about the quality of AI predictions. Such uncertainty is a recurring feature of many settings: models constantly change, new use cases emerge, and some use cases (e.g., scientific discovery) by nature demand using models in settings that are outside of the distribution of instances previously seen (Bommasani et al., 2021). It is often difficult for people to credibly know the joint distribution of model inputs and ground truth for the specific task they are solving (Vafa et al., 2024).

We study how asymmetric information about the quality of information available to a human decision maker vs an AI impacts the ability of a decision maker to extract complementary value from the AI. We start by introducing a formal model based on statistical decision theory in which the human does not know the exact joint distribution over AI predictions and ground truth. Instead, they have an *uncertainty set* which formalizes their beliefs about the model's quality and seek to ensure robust performance over that set. For example, a decision maker using AI likely believes that the AI has at least some level of correlation with the ground truth, even if the full joint distribution is unknown. We ask: what conditions must the decision maker be willing to impose about the performance of the AI in order to ensure that they benefit from using it?

It turns out that the presence of uncertainty substantially changes the answer. If the decision maker knows the true joint distribution, they benefit from using the AI if its prediction carries *any* signal in excess of what is already observed by the decision maker (essentially the condition tested in previous work). However, under uncertainty, it becomes much harder for the decision maker to construct a rule which makes nontrivial use of AI input and guarantees that the decision maker benefits compared to ignoring the AI entirely. A key factor turns out to be whether the errors in the AI's predictions are *negatively correlated* with the errors in the human's predictions. If this condition is satisfied, the decision maker can construct robust strategies which guarantee an improvement in expected utility. If it is not satisfied, the window for robust complementarity is much narrower: decisions to use the AI essentially reflect a weakness of the human's ability. In stylized terms, we interpret this as a movement towards automation instead of complementarity.

We then use empirical data from real-world forecasting and social science benchmarks that compare predictions from humans to those from LLMs. We compare human and LLM prediction errors and find that they are positively correlated,

[1]Machine Learning Department, Carnegie Mellon University. Correspondence to: Yewon Byun <yewonb@cs.cmu.edu>.

putting us in the "difficult" setting for complementarity. Alternative prompting strategies (e.g., instructing the LLM to focus on factors others may have missed) meaningfully reduce but fail to consistently eliminate positive correlations in errors. We suggest that explicitly optimizing for ability to provide complementary information should be a principle for LLM training and evaluation in the future, in order to ensure that they can be robustly used by humans to improve their own decision making.

## 2. Related Work

**Human-AI Decision Making.** A large literature examines joint decision making by humans and AI, often with a focus on when joint decision making can improve over either party individually ("complementarity"). Perhaps most related to our work, Steyvers et al. (2022) propose a generative model for combining human and AI confidences and show that the model theoretically implies conditions under which a human-AI team outperforms a team of multiple humans or multiple models. Our work differs in that we compare the human-AI combination to the single human alone, but focus on the role of uncertainty (where (Steyvers et al., 2022) assume in their theoretical analysis that the model parameters are known). Also related is (Donahue et al., 2022), who analyze a theoretical model of human decisions about when to rely on AI to identify conditions for complementarity, though without focusing on the role of uncertainty.

More broadly, there is a large algorithmic literature on structures for human-AI decision making, for example learning to defer (Madras et al., 2018; Mozannar & Sontag, 2020), triage decisions (Raghu et al., 2019; Okati et al., 2021), learning models that complement human decision makers (Wilder et al., 2020), and using AI to shape human choice sets (Straitouri et al., 2023). Nevertheless, a significant empirical literature typically fails to find evidence of complementarity between humans and AI (see Vaccaro et al. (2024) for a review). Our work theoretically explores one mechanism that could make complementarity more difficult to realize in real-world settings and explores its impact in real world data on human-LLM predictions.

**LLMs for Prediction.** A key motivation for our work is the growing interest in using LLMs to provide predictions that could augment human decision making. Since LLMs can be applied to new tasks in a zero-shot manner, humans are often faced with uncertainty about how a LLM will perform for a specific prediction task at hand. Existing work has begun to evaluate the performance of LLMs at forecasting future events (Karger et al., 2025; Yang et al., 2025; Ye et al., 2024; Halawi et al., 2024) and shown that they compare favorably to many humans. In scientific applications, several works have evaluated using LLMs to forecast the results of scientific experiments, for example in the social sciences (Hewitt et al., 2024; Park et al., 2024) or neuroscience (Luo et al., 2025). They similarly find that LLMs perform competitively with human forecasters. As a result, there is interest in using outputs from LLMs to "screen" potential experiments, using LLM predictions of effects as a sort of pilot study to decide which experiments to pursue (Anthis et al., 2025). This is the motivation for our "binary decisions" setting below, where we analyze conditions for LLMs to improve human decision making at such a task. We also use datasets from both forecasting and scientific settings to study the correlation between LLM and human errors and impact on complementary performance; existing work has focused mostly on comparing LLM and human accuracy and less on the relative distribution of errors between the two (though Hewitt et al. (2024) confirm that both human and AI information contain independent information predictive of true effects).

## 3. Model for Human-AI Complementarity

We assume that there is a ground-truth state of the world $\theta$ which the decision maker aims to infer. The decision maker observes a signal $\phi_H$ corresponding to their prior belief as well as a signal $\phi_{AI}$ generated by the machine learning system (e.g., a LLM). They then make a decision $d \in \mathcal{D}$. Their goal is to minimize a loss $\ell(d, \theta)$ which depends on the unknown state and on the decision.

We study a generative process in which the signals are noisy measurements of the ground truth state. We start with a stylized, linear-Gaussian version of the model which imposes normally distributed errors in the signals observed by each of the agents. This setting turns out to be sufficient to illustrate much of the intuition behind the problem structure, which we later show generalizes considerably. Formally, we start with the model

$$\begin{aligned}
\theta &\sim N(0, \sigma_\theta^2) \\
\phi_H &= \lambda_H \theta + \epsilon_H \\
\phi_{AI} &= \lambda_{AI} \theta + \epsilon_{AI} \\
\epsilon_H, \epsilon_{AI} &\sim N(0, \Sigma_\epsilon)
\end{aligned}$$

where $\epsilon_H$ and $\epsilon_{AI}$ are mean-zero Gaussian noise. This implies the joint distribution $[\theta \; \phi_H \; \phi_{AI}] \sim N(0, \Sigma)$. The core of the problem is encapsulated in the covariance matrix $\Sigma$. We start with this formalization and provide an essentially complete characterization when the decision maker provides a prediction of $\theta$ ($d \in \mathbb{R}$) and is evaluated via the mean-squared error $\ell(d, \theta) = (d - \theta)^2$. We then show that the qualitative conclusions generalize in two directions: a more complex decision making task in the Gaussian setting and predictions in a non-Gaussian generative model. Note that the main analysis assumes Gaussianity principally in the *errors* from the two signals: the human or AI may react in a complex, nonlinear fashion to whatever information they

observe, and we simply model the deviation between their predictions and the ground truth as normally distributed.

The expected loss under a particular distribution parameterized by $\Sigma$ is

$$L_\Sigma(d) = \mathbb{E}_{\theta, \phi_H, \phi_{AI} \sim N(0, \Sigma)}[\ell(d(\phi_H, \phi_{AI}), \theta)].$$

If the decision maker knew (or had samples from) the full joint distribution, they could simply minimize the expected loss over $d$. This is the setting discussed in a great deal of previous work about optimal human-AI decision making. The key feature of our formal model is to introduce uncertainty about the quality of the AI signal. In particular, our decision maker will stipulate that $\Sigma$ belongs to an uncertainty set $\mathcal{U}$ which pins down all features of the joint distribution *except* for the correlation of the AI signal with $\theta$.

**Assumption 3.1.** Every $\Sigma \in \mathcal{U}$ agrees on all entries except for $\mathrm{Cov}(\phi_{AI}, \theta)$.

As we will see, uncertainty about $\mathrm{Cov}(\phi_{AI}, \theta)$ is the core mechanism that generates a substantial difference in optimal policies. Additionally, in many settings, it may be reasonable to view the other entries of $\Sigma$ as known because it is cheap to generate samples of $\phi_{AI}$, which the decision maker can compare to their own belief; it is obtaining observations of the true state $\theta$ that is expensive. For example, a scientist can easily ask a language model for its forecast $\phi_{AI}$ about the results of many possible experiments, but the bottleneck is in actually performing experiments to observe $\theta$ and learn how accurate $\phi_{AI}$ really was. We expect that many of our conclusions (e.g., about the difficulty of achieving robust complementarity) would grow stronger if the model included uncertainty in other aspects of $\Sigma$ as well.

Our uncertainty-set formulation is compatible with richer models of uncertainty. For example, if the decision maker has a prior over possible AI signal qualities, model parameters, or data-generating processes, this uncertainty can be integrated out. From the perspective of the downstream decision problem, the relevant object is the induced predictive posterior over the state given the observed human and AI signals. Thus, uncertainty over intermediate quantities affects the analysis only through the posterior distribution it induces. When this induced posterior satisfies the structural conditions assumed in our analysis—for example, the Gaussian covariance restrictions in Section 3, or the finite-moment and negative-dependence conditions in Section 3.2—the corresponding results apply without further modification.

The same observation applies to estimation error in the human signal or in the joint distribution of human and AI signals. If realizations of the state are costly, the decision maker may place a prior over these quantities and update after observing samples of the true state. After this update, the decision-relevant object is again the post-update predic-

tive posterior. Our framework can therefore be viewed as placing a credal set over plausible predictive posteriors and asking when complementarity is guaranteed uniformly over that set.

In order to restrict to more interesting cases, we will also impose the assumption that neither agent observes a noiseless measurement of $\theta$:

**Assumption 3.2.** $\mathrm{Var}(\phi_H | \theta)$ and $\mathrm{Var}(\phi_{AI} | \theta)$ are both strictly greater than 0.

Finally, we will impose without loss of generality that $\phi_H$ is scaled so that $\lambda_H = \frac{\mathrm{Cov}(\theta, \phi_H)}{\mathrm{Var}(\phi_H)} = 1$, i.e., the linear regression coefficient of $\theta$ on $\phi_H$ is 1. This is simply a normalization that does not further restrict the set of $\mathcal{U}$ under consideration.

We study: under what conditions is there a rule $d$ which makes nontrivial use of $\phi_{AI}$ and which guarantees better value than one which uses only $\phi_H$? Let $d_H$ denote the optimal decision rule which is a function only of $\phi_H$. Since the class of uncertainty sets we consider fully specify the $(\theta, \phi_H)$ distribution, $d_H$ can be characterized explicitly for many decision problems. We study when it is possible to exhibit another *joint* decision rule $d_J$ which is a function of $\phi_{AI}$ as well as $\phi_H$ and which satisfies

$$L_\Sigma(d_J) \leq L_\Sigma(d_H) \quad \forall \Sigma \in \mathcal{U}$$

and there exists at least one $\Sigma \in \mathcal{U}$ for which the inequality is strict.

We start by exploring this question for prediction under mean squared error. Here, the action taken by the decision maker is to produce a prediction $d \in \mathbb{R}$ which is evaluated according to the loss $\ell(d, \theta) = (d - \theta)^2$. For example, this setting could model a forecaster who seeks to make accurate predictions about an unknown event, as in efforts to use LLMs for forecasting (where MSE, aka Brier score, is a common metric).

As a starting point, we show that it suffices to confine ourselves to decision rules that can be represented as linear functions $d = a\phi_H + b\phi_{AI}$. This is intuitive because conditional expectations are linear in the signals under joint normality. The only subtlety is ensuring that the desire for robust performance simultaneously across many distributions does not create the need for nonlinear decision rules. We prove that this is not the case:

**Proposition 3.3.** *Given any decision rule $d$, there exists a linear decision rule $\tilde{d}$ such that $L_\Sigma(\tilde{d}) \leq L_\Sigma(d)$ for all $\Sigma \in \mathcal{U}$.*

Further, our normalization ensures that $a = 1$ in the optimal decision rule, i.e., without any additional information, the decision maker simply forecasts $d_H(\phi_H) = \phi_H$ as their prediction of $\theta$. We are interested in conditions under which

there exists a decision rule $d_b(\phi_H, \phi_{AI}) = \phi_H + b\phi_{AI}$ for $b \neq 0$ which provably dominates $d_H$ across $\mathcal{U}$. We show that the following two conditions suffice:

**Proposition 3.4.** *Suppose that all $\Sigma \in \mathcal{U}$ satisfy the following two additional conditions: (1) $Cov(\phi_{AI}, \theta) > \delta$ for some $\delta > 0$. (2) $Cov(\phi_H, \phi_{AI}|\theta) \leq 0$. Then, there exists a $b \neq 0$ such that $d_b$ strictly dominates $d_H$ over $\mathcal{U}$.*

Intuitively, (1) is a minimal belief for the decision maker to hold: to rationalize using the machine signal, they must believe that it is positively correlated with $\theta$ by at least some amount. (2) is equivalent to $\epsilon_H$ and $\epsilon_{AI}$ being negatively correlated, i.e., negative correlation in the errors.

We now provide the intuition behind the construction of the decision rule, which will also underlie results for the generalized settings discussed in later sections. Define the AI residual $r$ by regressing out the portion of $\phi_{AI}$ predictable from $\phi_H$

$$r \triangleq \phi_{AI} - \frac{\mathrm{Cov}(\phi_H, \phi_{AI})}{\mathrm{Var}(\phi_H)} \phi_H. \tag{1}$$

By construction, $r$ now satisfies $\mathrm{Cov}(\phi_H, r) = 0$. For each $\Sigma \in \mathcal{U}$, define the regression coefficient of $r$ on $\theta$:

$$\beta(\Sigma) \triangleq \frac{\mathrm{Cov}_\Sigma(r, \theta)}{\mathrm{Var}(r)}, \tag{2}$$

We can show that the regression coefficient is strictly positive under the same hypotheses for Proposition 3.4 (positive marginal correlation of $\phi_{AI}$ combined with negative correlation of errors):

**Lemma 3.5.** *Under the conditions of Proposition 3.4, for all $\Sigma \in \mathcal{U}$, $\beta(\Sigma) \geq b = \delta \frac{Var(\phi_H|\theta)}{Var(\phi_H)Var(r)} > 0$,*

Effectively, the strictly improving decision rule corresponds to the prediction $\phi_H + b \cdot r$. Since the true regression coefficient under any $\Sigma \in \mathcal{U}$ is guaranteed to be at least $b$, this corresponds to a conservative counterpart of the ideal linear regression of $\theta$ on $(\phi_{AI}, \phi_H)$, which is guaranteed to improve performance in mean squared error.

We then ask whether a weaker version of Condition 2 suffices. For example, what if the errors are only modestly positively correlated? We give an exact characterization, showing that in the positive correlation regime, the existence of an improving decision rule requires that the decision maker's error $\epsilon_H$ grow in direct proportion to the amount of positive correlation:

**Proposition 3.6.** *Define the extremal values*

$$\delta := \inf_{\Sigma \in \mathcal{U}} Cov_\Sigma(\theta, \phi_{AI}), \qquad \gamma := \sup_{\Sigma \in \mathcal{U}} Cov_\Sigma(\phi_H, \phi_{AI} \mid \theta).$$

*Then there exists a coefficient $b^\star > 0$ such that $d_{b^\star}$ strictly dominates $d_H$ over $\mathcal{U}$ if and only if*

$$\gamma < \delta \, \frac{Var(\phi_H \mid \theta)}{Var(\phi_H)}.$$

*The maximal possible improvement in loss for any $d_b$ over $d_H$, which is guaranteed for every $\Sigma \in \mathcal{U}$ is given by*

$$\frac{\left( \delta \, \frac{Var(\phi_H|\theta)}{Var(\phi_H)} - \gamma \right)^2}{Var(\phi_{AI})}.$$

Intuitively, any positive correlation in errors ($\gamma$) must be matched by a combination of higher absolute performance for the AI signal ($\delta$) and noise in the human signal ($\mathrm{Var}(\phi_H|\theta)$). Rationalizing nontrivial dependence on the AI signal in the positive correlation case thus requires that the AI perform better and the human perform worse in direct proportion to the amount of positive correlation.

By contrast, no such dilemma appears without the presence of uncertainty. For any fixed $\Sigma$, the decision maker can benefit from both signals whenever the information contained in $\phi_{AI}$ is not completely redundant with $\phi_H$:

**Proposition 3.7.** *For any fixed covariance structure $\Sigma$, so long as $\beta(\Sigma) \neq 0$, there is a joint decision rule $d_b$, $b \neq 0$, which satisfies $L_\Sigma(d_b) < L_\Sigma(d_H)$.*

Effectively, the presence of uncertainty about the AI signal significantly shrinks the region for complementarity: instead of any new information being useful, the decision maker requires enough structure to guarantee that the new information contributed by the AI (the residual) is correlated in a known way with the ground truth $\theta$.

### 3.1. Binary Decisions

We next study a setting in which a decision maker takes a binary action $d \in \{0, 1\}$ after observing signals. The interpretation is that $d = 1$ corresponds to *investing* in a project (e.g. pursuing an experiment), while $d = 0$ corresponds to not investing. The state of the world is $\theta$, and investing is desirable precisely when $\theta$ exceeds a fixed quality threshold $\tau$. Investing incurs a cost $c \in (0, 1)$, while a successful investment yields a unit payoff (without loss of generality). Thus the decision maker's utility is

$$u(d, \theta) \triangleq d \cdot 1\{\theta \geq \tau\} - c \cdot d, \tag{3}$$

and the corresponding loss is $\ell(d, \theta) = -u(d, \theta)$. Because $d$ is binary, the optimal action is to invest if and only if the posterior success probability exceeds the cost:

$$\mathbb{P}(\theta \geq \tau) \geq c.$$

In particular, under our standing normalization $\mathbb{E}[\theta \mid \phi_H] = \phi_H$ and $\mathrm{Var}(\theta \mid \phi_H) = \sigma_H^2$, the expert-only posterior success probability is, for every $\Sigma \in \mathcal{U}$,

$$p_H(x) \triangleq \mathbb{P}(\theta \geq \tau \mid \phi_H = x) = \Phi\left(\frac{x - \tau}{\sigma_H}\right),$$

which is strictly increasing in $x$. Let $x_H$ be the unique threshold satisfying $p_H(x_H) = c$, and define the human-only optimal decision rule

$$d_H(\phi_H) \triangleq 1\{\phi_H \geq x_H\}. \tag{4}$$

Our goal is to construct a decision rule that uses both signals $(\phi_H, \phi_{AI})$ and robustly improves upon $d_H$ over the uncertainty set $\mathcal{U}$. The starting point is the same construction used in the previous section, based on the residual AI signal and its regression coefficient $\beta(\Sigma)$ for $\theta$ under a given distribution $\Sigma$. Lemma 3.5 lower bounds the value of $\beta(\Sigma)$ under negative correlation of errors. Our strategy will be to use this lower bound on the regression coefficient to form a "pessimistic" estimate of the mean and variance of $\theta$ conditional on $(\phi_H, \phi_{AI})$. Define

$$\mu_{\mathrm{b}}(\phi_H, r) \triangleq \phi_H + br, \qquad \sigma_b^2 \triangleq \sigma_H^2 - b^2 \mathrm{Var}(r), \tag{5}$$

While it would be tempting to simply construct an analogy to the human-only rule $d_H$ using $\mu_{\mathrm{b}}$ and $\sigma_{\mathrm{b}}$, this neglects the fact that decision-making depends on the entire distribution of $\theta$ in this setting, not only on its mean as before. For example, it is possible for observing a positive residual to increase the posterior mean of $\theta$ but actually decrease the posterior $\mathrm{Pr}(\theta \geq \tau)$. To circumvent this, we define conservative bounds on the success probability which are guaranteed to hold across the entirety of $\mathcal{U}$:

$$P_{\mathrm{low}}(x, y) \triangleq \begin{cases} \Phi\left(\dfrac{\mu_{\mathrm{b}}(x, y) - \tau}{\sigma_b}\right) & y \geq 0 \wedge \mu_{\mathrm{b}}(x, y) \geq \tau \\ 0, & \text{otherwise,} \end{cases}$$

$$P_{\mathrm{high}}(x, y) \triangleq \begin{cases} \Phi\left(\dfrac{\mu_{\mathrm{b}}(x, y) - \tau}{\sigma_H}\right) & \mu_{\mathrm{b}}(x, y) < \tau \wedge y < 0 \\ 1, & \text{otherwise.} \end{cases}$$

We will use these conservative bounds on the success probability to construct a new rule that overrules the human-only decision rule whenever doing so is guaranteed to improve across the entirety of $\mathcal{U}$. Formally we define

$$d_J(\phi_H, \phi_{AI}) \triangleq \begin{cases} 1, & \text{if } P_{\mathrm{low}}(\phi_H, r) \geq c, \\ 0, & \text{if } P_{\mathrm{high}}(\phi_H, r) \leq c, \\ d_H(\phi_H), & \text{otherwise,} \end{cases} \tag{6}$$

**Proposition 3.8.** *Under the same assumptions as Proposition 3.4, the joint rule $d_J$ weakly dominates the expert-only optimal rule $d_H$ in expected utility:*

$$\mathbb{E}_\Sigma\big[u(d_J(\phi_H, \phi_{AI}), \theta)\big] \geq \mathbb{E}_\Sigma\big[u(d_H(\phi_H), \theta)\big] \ \forall \Sigma \in \mathcal{U}.$$

*Moreover, the inequality is strict for any $\Sigma \in \mathcal{U}$ such that $\beta(\Sigma) > b$.*

This shows that the basic conclusion from the mean squared error setting generalizes to a more complex problem with binary decisions: joint decision rules can ensure robust improvements in utility when errors between the AI and human signals are negatively correlated. We provide an analysis for investment decisions in the case of positive correlations in Appendix A.

### 3.2. Beyond the Gaussian model

We finally show that the core results on conditions for improvement in mean squared error extend beyond the linear-Gaussian model to a substantially more general class of *nonlinear but monotone* signal structures which do not assume joint normality. We generalize the model as follows. Let $\theta$ be a real-valued random variable with $\mathbb{E}[\theta] = 0$ and $\mathrm{Var}(\theta) < \infty$. The signals follow the generative model

$$\phi_H = f(\theta) + \varepsilon_H, \qquad \phi_{AI} = g(\theta) + \varepsilon_{AI},$$

where $f, g : \mathbb{R} \to \mathbb{R}$ are strictly increasing and continuously differentiable and $\epsilon_H$ and $\epsilon_{AI}$ are random variables with expectation 0. We assume that $(\epsilon_H, \epsilon_{AI}) \perp \theta$ and that both have finite second moments. Monotonicity of $f$ and $g$ implies that each agent's signal will increase in expectation with the ground truth state, an analogue of positive correlation in the Gaussian state. The joint distribution $(\theta, \phi_H, \phi_{AI})$ is fully specified by the tuple $\xi = (f, g, P(\theta), P(\epsilon_H, \epsilon_{AI}))$. An uncertainty set $\mathcal{U}$ is thus a set of potential values for $\xi$. In analogy to the uncertainty sets defined earlier for the Gaussian case, we will consider $\mathcal{U}$ for which all $\xi \in \mathcal{U}$ agree on the distributions $P(\theta, \phi_H)$ and $P(\phi_H, \phi_{AI})$. Uncertainty lies in the relationship between the AI signal and the truth, reflected in $g$ and the distribution of $\epsilon_{AI}$.

We study conditions in this model where a decision rule that uses $\phi_{AI}$ has robustly lower mean squared error. We show that conditions analogous to those in the Gaussian case suffice: the AI signal must be sufficiently informative about $\theta$, and the errors must be negatively dependent. To formalize the first condition, we assume that the AI signal has a uniformly bounded-below derivative:

$$g'(t) \geq k > 0 \quad \text{for all } t \in \mathbb{R}. \tag{7}$$

This captures the idea that the AI is *uniformly sensitive* to changes in $\theta$, in the sense that increasing $\theta$ always pushes $\phi_{AI}$ up at least at rate $k$.

To formalize the idea of negative dependence in errors, we consider the condition that

$$\mathbb{E}[\varepsilon_{AI} \mid \varepsilon_H] \text{ is nonincreasing in } \varepsilon_H \text{ almost surely,} \tag{8}$$

so that errors in the positive direction for the human imply that the AI will, on average, have smaller errors. This

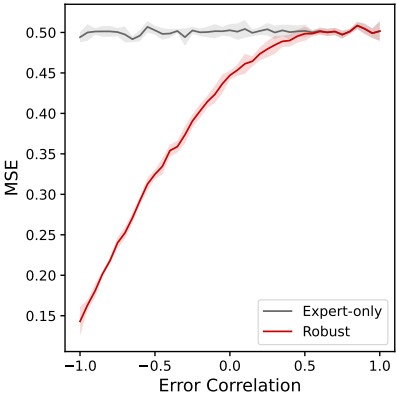

*Figure 1.* MSE of the expert-only estimator and our robust estimator as we vary the correlation in expert and LLM errors (lower is better). The MSE of our estimator strongly dominates the expert-only baseline when errors are negatively correlated.

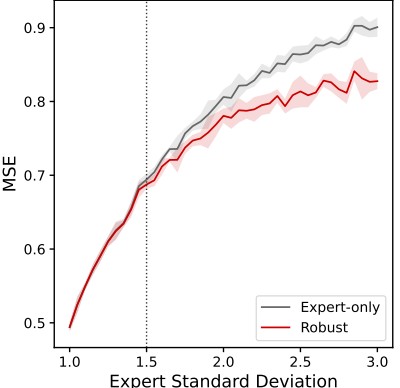

*Figure 2.* MSE of the expert-only estimator and our robust estimator as we vary the expert uncertainty under the positive correlation setting. Our robust estimator begins to dominate the expert-only baseline when human uncertainty exceeds that of the AI signal. The vertical dotted line denotes the AI signal standard deviation.

condition is satisfied by negative conditional covariance in the Gaussian case but now allows us to extend the results without distributional assumptions.

The core of the strategy we use to construct a loss-improving decision rule is the *nonlinear residual* of the AI signal after conditioning on the human's signal:

$$\tilde{r}(\phi_H, \phi_{AI}) \triangleq \phi_{AI} - \mathbb{E}[\phi_{AI} \mid \phi_H]. \tag{9}$$

Our goal is to show that under the structural assumptions above, $\tilde{r}$ is *still positively correlated with* $\theta$, and that this can be uniformly exploited to improve squared-error risk over any predictor that uses only $\phi_H$.

**Proposition 3.9.** *If Equations 7 and 8 hold for all $\xi \in \mathcal{U}$, $\tilde{r}$ satisfies*

$$Cov(\tilde{r}, \theta) \geq k \, \mathbb{E}\big[Var(\theta \mid \phi_H)\big].$$

*for all $\xi \in \mathcal{U}$ as well.*

When the residual is guaranteed to have positive correlation with $\theta$, we can construct a decision rule using a strategy very similar to the Gaussian case: start with the regression of $\theta$ on $\phi_H$ and then add $b \cdot \tilde{r}$ for an appropriately chosen coefficient $b$.

**Proposition 3.10.** *Consider the optimal human-only decision rule $d_H(\phi_H) = \mathbb{E}[\theta|\phi_H]$ and the class of joint decision rules $d_b(\phi_H, \phi_{AI}) = \mathbb{E}[\theta|\phi_H] + b \cdot \tilde{r}(\phi_H, \phi_{AI})$. Whenever the conditions of Equations 7 and 8 hold for all $\xi \in \mathcal{U}$, and $E[Var(\theta \mid \phi_H)] > 0$, there exists a $b > 0$ such that $L_\xi(d_b) < L_\xi(d_H)$ for all $\xi \in \mathcal{U}$.*

That is, negative dependence in errors is sufficient to ensure robust complementarity under mean squared error in a much broader class of non-Gaussian settings.

## 4. Synthetic Experiments

We start by using simulation to illustrate how the performance of the robust decision rules constructed above compares to human-only decision making as the joint distribution of errors varies. We simulate data from the Gaussian measurement model, which allows us to directly control the error correlation structure by modifying the covariance matrix between human and AI signals. In the synthetic experiments where we vary $\rho$, the human signal is slightly more accurate than the AI signal. We defer specific details about the dataset to Appendix C.2.

### 4.1. Synthetic Experiments for MSE

**Varying human and AI error correlations.** Figure 1 reports the MSE of both estimators as a function of the error correlation $\rho$. Consistent with our theoretical analysis, the robust estimator yields the largest gains when errors are negatively correlated. As $\rho$ becomes large, the improvement over the human-only baseline diminishes: despite the fact that the AI signal contains useful information, it becomes difficult to robustly benefit from.

**Varying human uncertainty under positive correlation.** We next fix the errors to be positively correlated ($\rho = 0.75$) while varying the expert uncertainty. Figure 2 shows that under positive correlation, the robust estimator outperforms the expert-only baseline only when the uncertainty in the human signal exceeds that of the AI signal, with significant gains limited to the case where the human signal is highly noisy.

### 4.2. Synthetic Experiments for Investment Decisions

**Varying human and AI error correlations.** Figure 3 reports the average utility as a function of the error correlation

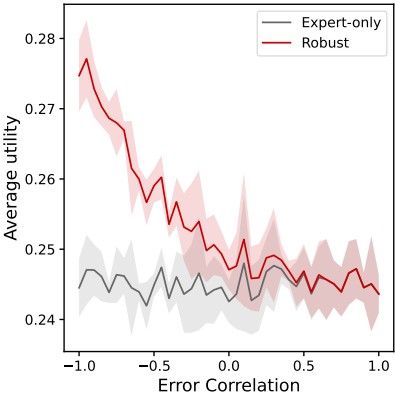

*Figure 3.* Average utility of the expert-only investment policy $d_H$ and the robust symmetric policy $d_{\mathrm{sym}}$ as we vary the correlation $\rho$ between human and AI errors (higher is better). Shaded regions indicate $\pm 1$ standard deviation across seeds.

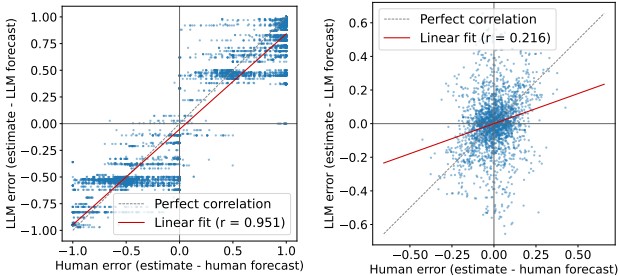

*Figure 4.* Correlation of errors of LLM forecasts vs. human forecasts on ForecastBench (top) and TESS studies (bottom). We find that **errors are positively correlated**, which is the difficult setting for complementarity.

$\rho$. Consistent with our theory, the robust symmetric policy yields the largest gains when errors are negatively correlated; in this regime, the residual $r$ is informative about the human residual uncertainty, and incorporating it improves decisions relative to $d_H$. As $\rho$ becomes large, the residualized machine signal becomes less complementary, and the robust policy reverts to the expert-only rule.

**Varying human uncertainty under positive correlation.** We next fix a positive correlation level ($\rho = 0.75$) while varying the expert uncertainty. Figure 5 (deferred to Appendix C.1) shows that when the human signal is already precise (small $\sigma_h$), the robust procedure matches the human-only policy, reflecting limited complementarity from the AI under strong positive correlation. As the human signal becomes noisier, the incremental information in the residualized AI signal becomes more valuable, and $d_{\mathrm{sym}}$ begins to dominate $d_H$ in average utility.

## 5. Real World Experiments

**Datasets and Models.** We conduct experiments on two real-world forecasting benchmarks. The first is Forecast-

Bench (Karger et al., 2025), which provides human forecasts (n=7,383) for prediction questions drawn from time series datasets and prediction market platforms. The second is a suite of 43 different social science survey experiments conducted via the Time-Sharing Experiments for the Social Sciences (TESS) project (Time-sharing Experiments for the Social Sciences) between 2016 and 2022, for which Hewitt et al. (2024) collected human and LLM forecasts of effect sizes. We conduct all experiments with GPT-5 and GPT-5-mini (Singh et al., 2025).

### 5.1. Real-World Empirical Results

We ask whether current models naturally satisfy conditions on the error structure in real-world forecasting experiments. In Figure 4, we find that contrary to this requirement, human and LLM errors are positively correlated, quite strongly for ForecastBench (Karger et al., 2025) and more modestly (but still non-trivially) for the TESS studies. This raises the question of whether such patterns are easily modifiable by changes in prompting to the model, or whether deeper interventions would be required to surface complementary information.

### 5.2. Potential Mitigation Strategies

We investigate whether post-hoc prompting strategies can induce negative correlation in prediction errors by encouraging: (i) explicit prediction and correction of human errors; (ii) divergent or contrarian reasoning, either relative to provided human rationales or to the model's own initial reasoning traces; and (iii) reliance on different information sets, forcing the model to attend to different signals. Details of each prompt strategy are provided in Appendix C.4 (see Figures 6 and 7).

Critically, while a shift in correlation is evident for several strategies, we observe only a single instance (out of 12 combinations of prompting strategies × tasks) in which correlation shifts from positive to negative (see Figures 6 and 7). This suggests that enforcing the error structures required for complementarity via prompting alone is nontrivial and unreliable. More broadly, these results suggest that post-hoc interventions are unlikely to consistently induce the error structure necessary for robust human-AI complementarity. This highlights the necessity for new training methods or objectives to optimize for outputs in this manner, when predictions will be used to complement human signals.

In Table 1, we observe that very few prompting strategies lead to gains of the combined robust MSE over the optimal human MSE on the forecasting task. This is explained by the fact that all cases remain in the positively correlated error regime, where our theory expects to have no or little improvement, with the sole exception of Contrarian Reasoning on TESS. In this particular case, we see a slight

*Table 1.* We report the MSE and correlation structure resulting from various prompting strategies. Human MSE denotes the human signal MSE. LLM MSE denotes the LLM signal MSE. Calibrated Human MSE denotes the optimal predictor's MSE based on the human signal. $\rho(\epsilon_H, \epsilon_{AI})$ denotes the Pearson correlation coefficient over the human error and the LLM error. Robust MSE represents the performance of the robust combination of the LLM and human signal. Note that human rationale and additional context fields are not available for the TESS studies. Therefore, we only run the Human-Divergent Reasoning and Restricted Context strategies for ForecastBench.

| Dataset/Prompting Strategy | Human MSE | LLM MSE | Calibrated Human MSE | $\rho(\epsilon_H, \epsilon_{AI})$ | Robust MSE |
|---|---|---|---|---|---|
| **ForecastBench** | | | | | |
| Baseline | 0.514 | 0.451 | 0.154 | 0.963 | 0.154 |
| Direct Error Correction | 0.514 | 0.506 | 0.154 | 0.995 | 0.154 |
| Contrarian Reasoning | 0.514 | 0.216 | 0.154 | 0.498 | 0.153 |
| Self-Divergent Reasoning | 0.514 | 0.461 | 0.154 | 0.966 | 0.154 |
| Human-Divergent Reasoning | 0.514 | 0.453 | 0.154 | 0.985 | 0.154 |
| Restricted Context | 0.514 | 0.439 | 0.154 | 0.967 | 0.154 |
| **TESS Studies** | | | | | |
| Baseline | 0.0104 | 0.0285 | 0.00524 | 0.216 | 0.00510 |
| Direct Error Correction | 0.0104 | 0.0230 | 0.00524 | 0.175 | 0.00520 |
| Contrarian Reasoning | 0.0104 | 0.0351 | 0.00524 | -0.338 | 0.00490 |
| Self-Divergent Reasoning | 0.0104 | 0.0185 | 0.00524 | 0.213 | 0.00519 |

*Table 2.* **Distribution shift on ForecastBench and TESS.** Mean squared error (MSE) on distribution shift (i.e., generalizing to a new topic) and in-distribution (i.e., training on the same topic and evaluating on a test split). MSEs on TESS are reported in units of $10^{-3}$. Values are mean $\pm$ std across random splits. $\Delta$ is OOD MSE - Target Domain ID MSE (positive indicates degradation under shift). Rightmost column reports the held-out signal-coupling diagnostic $\mathrm{Cov}(\phi_H, \phi_{AI})/\mathrm{Var}(\phi_{AI})$.

| Dataset | Topic (Keywords)/Group | OOD MSE | Target Domain ID MSE | $\Delta$ | $\dfrac{\mathrm{Cov}(\phi_H, \phi_{AI})}{\mathrm{Var}(\phi_{AI})}$ |
|---|---|---|---|---|---|
| ForecastBench | (data, series) | $0.152 \pm 0.041$ | $0.182 \pm 0.066$ | -0.031 | 0.802 |
| | (price, close) | $0.178 \pm 0.012$ | $0.343 \pm 0.151$ | -0.165 | -0.137 |
| | (violence, events) | $0.121 \pm 0.029$ | $0.111 \pm 0.041$ | +0.010 | 0.900 |
| | (market, prediction) | $0.170 \pm 0.045$ | $0.137 \pm 0.035$ | +0.032 | 0.769 |
| TESS Studies | agenda setting | $2.76 \pm 0.46$ | $2.42 \pm 0.34$ | +0.34 | 0.42 |
| | framing | $9.31 \pm 1.05$ | $9.79 \pm 1.02$ | -0.49 | 0.79 |
| | gender norms | $9.96 \pm 2.92$ | $11.05 \pm 2.69$ | -1.09 | 0.44 |
| | immigration attitudes | $0.96 \pm 0.21$ | $0.63 \pm 0.07$ | +0.33 | 2.03 |
| | international rights | $2.46 \pm 0.19$ | $1.88 \pm 0.10$ | +0.58 | 0.67 |
| | partisanship polarization | $0.50 \pm 0.09$ | $0.46 \pm 0.09$ | +0.05 | 0.54 |
| | race ethnicity | $1.87 \pm 0.26$ | $0.36 \pm 0.06$ | +1.50 | 0.94 |
| | representation | $0.38 \pm 0.30$ | $0.06 \pm 0.02$ | +0.33 | 1.58 |
| | sexual misconduct credibility | $14.05 \pm 2.35$ | $8.69 \pm 0.66$ | +5.37 | 0.29 |
| | transgender rights | $1.81 \pm 0.38$ | $0.84 \pm 0.22$ | +0.97 | 1.25 |

improvement in MSE for our robust combination estimator. We note that improvements are small as the human signal is substantially stronger than the LLM signal (optimal fit MSE 0.0052 vs. 0.0206).

### 5.3. Distribution Shift Setting

A natural question is whether the covariance structure between expert and model predictions, once estimated, can be reused for future prediction tasks. For instance, having expended the cost to collect forecasts on a certain domain, one might hope to learn the empirical relationship between human and LLM prediction errors and then apply it to new studies. However, this assumes that the expert-model relationship is stable across settings. This is especially relevant in settings, such as scientific discovery, where researchers

study novel phenomena and the degree to which a specific area constitutes a coherent domain with stable prediction error correlations is unclear. Even within a single field, the relationship between expert and model signals may shift as the nature of the forecasting task changes.

We investigate this via leave-one-topic-out evaluation, evaluating performance of models to generalize out-of-distribution across topics. For ForecastBench, we induce topic clusters using BERTopic (Grootendorst, 2022); for TESS, we use the native study categories (e.g., framing, immigration attitudes, gender norms). In each fold, we fit a linear combination $\hat{\theta} = \beta_0 + \beta_1 \phi_H + \beta_2 \phi_{AI}$ on in-distribution data and evaluate on the held-out topic. We report the estimation error $(\theta_B - \hat{\theta}_B)$. We also compare $\frac{Cov(\phi_H, \phi_{AI})}{Var(\phi_{AI})}$ across settings, which characterizes how coupled the expert

and AI signals are.

The results reveal substantial heterogeneity (Table 2). Some topics transfer well: on ForecastBench, "price/close" questions show $\Delta = -0.165$, meaning the out-of-distribution estimator actually performs better (perhaps due to more training data). Others fail dramatically: on TESS, "sexual misconduct credibility" exhibits $\Delta = +5.37 \times 10^{-3}$, a large increase in MSE. Across both datasets, the largest failures occur when the diagnostic $\mathrm{Cov}(\phi_H, \phi_M)/\mathrm{Var}(\phi_M)$ diverges most from the source domain, or where there are changes in the relationship between human and model signals rather than covariate shift alone. This suggests that it can be used as a proxy, which does not require any ground truth, to determine whether there is potential failure due to distribution shift. These findings complicate naive deployment of models based on human-AI collaboration across different scientific topics. The error structure that determines complementarity varies with the task distribution in ways that are difficult to anticipate.

## 6. Conclusion

In this work, we take a step towards understanding the conditions under which human-AI collaboration provably improves upon either agent alone. Our theoretical analysis reveals that complementarity is not guaranteed by predictive accuracy alone, but that the error correlation structure between human and model predictions plays a significant role in determining whether collaboration yields better performance. We derive precise conditions on this structure that must be satisfied for complementary gains to emerge, and show that these conditions can be surprisingly stringent in practice. Empirically, we find that current models do not naturally satisfy these conditions across a range of real-world forecasting benchmarks. Simple post-hoc mitigation strategies–including prompting interventions designed to induce negative error correlation–show limited effectiveness in enforcing the required error structure. These findings suggest that achieving genuine complementarity may require more fundamental interventions, potentially in earlier stages of the training paradigm. Our work highlights an important gap between the promise of human-AI collaboration and its current realization. We hope these theoretical and empirical observations motivate further investigation into strategies that explicitly optimize for and prioritize complementarity.

## Impact Statement

This work characterizes the conditions under which human-AI collaboration yields robust improvements, aiming to support more informed deployment of AI systems in consequential domains. Our framework provides decision makers with principled guidance on when and how AI predictions can complement human judgment, rather than simply re-

place it. More broadly, we hope this work motivates further investigation into methods that prioritize complementarity.

## Acknowledgments

This material is based upon work supported by the AI Research Institutes Program funded by the National Science Foundation under AI Institute for Societal Decision Making (AI-SDM), Award No. 2229881.

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

## A. Positive Correlation Analysis for Investment Decisions

We now present an analysis of the case of positively correlated errors for the investment decisions setting.

**Proposition A.1.** *Define the extremal values*

$$\delta \triangleq \inf_{\Sigma \in \mathcal{U}} s(\Sigma), \qquad \gamma \triangleq \sup_{\Sigma \in \mathcal{U}} Cov_\Sigma(\phi_H, \phi_{AI} \mid \theta),$$

*where $s(\Sigma) = Cov_\Sigma(\theta, \phi_{AI})$. Assume the strict inequality*

$$\gamma \; < \; \delta \, \frac{Var(\phi_H \mid \theta)}{Var(\phi_H)}.$$

*Define the positive-correlation margin and the corresponding pessimistic regression lower bound*

$$\kappa_\gamma \; \triangleq \; \delta \, \frac{Var(\phi_H \mid \theta)}{Var(\phi_H)} - \gamma, \qquad b_{\mathrm{pos}} \; \triangleq \; \frac{\kappa_\gamma}{Var(r)} \; > \; 0.$$

*Then $\beta(\Sigma) \geq b_{\mathrm{pos}}$ for all $\Sigma \in \mathcal{U}$. Consequently, the symmetric robust investment rule $d_J$ constructed using $b = b_{\mathrm{pos}}$ weakly dominates the expert-only rule $d_H$ in expected utility:*

$$\mathbb{E}_\Sigma[u(d_J(\phi_H, \phi_{AI}), \theta)] \; \geq \; \mathbb{E}_\Sigma[u(d_H(\phi_H), \theta)] \qquad \forall \Sigma \in \mathcal{U}.$$

*Moreover, the inequality is strict for any $\Sigma \in \mathcal{U}$ for which $\beta(\Sigma) > b_{\mathrm{pos}}$ (on a set of positive probability in $(\phi_H, r)$).*

*Proof.* Fix $\Sigma \in \mathcal{U}$. By the definition of $r$ and the normalization $\mathrm{Cov}(\theta, \phi_H) = \mathrm{Var}(\phi_H)$,

$$\mathrm{Cov}_\Sigma(r, \theta) = \mathrm{Cov}_\Sigma(\phi_{AI}, \theta) - \mathrm{Cov}(\phi_H, \phi_{AI}).$$

Under joint Gaussianity, the law of total covariance yields

$$\mathrm{Cov}(\phi_H, \phi_{AI}) = \mathrm{Cov}_\Sigma(\phi_H, \phi_{AI} \mid \theta) + \frac{\mathrm{Cov}(\phi_H, \theta) \, \mathrm{Cov}_\Sigma(\phi_{AI}, \theta)}{\mathrm{Var}(\theta)} = \mathrm{Cov}_\Sigma(\phi_H, \phi_{AI} \mid \theta) + \frac{\mathrm{Var}(\phi_H) \, \mathrm{Cov}_\Sigma(\phi_{AI}, \theta)}{\mathrm{Var}(\theta)}.$$

Substituting and simplifying gives

$$\mathrm{Cov}_\Sigma(r, \theta) = \mathrm{Cov}_\Sigma(\phi_{AI}, \theta) \Big( 1 - \frac{\mathrm{Var}(\phi_H)}{\mathrm{Var}(\theta)} \Big) - \mathrm{Cov}_\Sigma(\phi_H, \phi_{AI} \mid \theta).$$

Finally, the conditional-variance identity and $\mathrm{Cov}(\phi_H, \theta) = \mathrm{Var}(\phi_H)$ imply

$$\mathrm{Var}(\phi_H \mid \theta) = \mathrm{Var}(\phi_H) - \frac{\mathrm{Cov}(\phi_H, \theta)^2}{\mathrm{Var}(\theta)} = \mathrm{Var}(\phi_H) \Big( 1 - \frac{\mathrm{Var}(\phi_H)}{\mathrm{Var}(\theta)} \Big),$$

so $\big( 1 - \mathrm{Var}(\phi_H)/\mathrm{Var}(\theta) \big) = \mathrm{Var}(\phi_H \mid \theta)/\mathrm{Var}(\phi_H)$. Hence

$$\mathrm{Cov}_\Sigma(r, \theta) = \mathrm{Cov}_\Sigma(\phi_{AI}, \theta) \frac{\mathrm{Var}(\phi_H \mid \theta)}{\mathrm{Var}(\phi_H)} - \mathrm{Cov}_\Sigma(\phi_H, \phi_{AI} \mid \theta) \; \geq \; \delta \frac{\mathrm{Var}(\phi_H \mid \theta)}{\mathrm{Var}(\phi_H)} - \gamma = \kappa_\gamma.$$

Dividing by $\mathrm{Var}(r)$ yields $\beta(\Sigma) = \mathrm{Cov}_\Sigma(r, \theta)/\mathrm{Var}(r) \geq \kappa_\gamma/\mathrm{Var}(r) = b_{\mathrm{pos}}$. With this uniform lower bound in hand, the dominance argument for the symmetric rule $d_J$ (given conservative bounds $P_{\mathrm{low}}$ and $P_{\mathrm{high}}$) applies verbatim, proving $\mathbb{E}_\Sigma[u(d_J, \theta)] \geq \mathbb{E}_\Sigma[u(d_H, \theta)]$ for all $\Sigma \in \mathcal{U}$. Strictness holds whenever $\beta(\Sigma) > b_{\mathrm{pos}}$ on a set of positive probability. $\qquad \square$

In the decision-making setting, robust complementarity for investment decisions again hinges on the AI signal being informative about the *human residual* rather than merely replicating the expert's mistakes. A similar condition $\kappa_\gamma = \delta \frac{\mathrm{Var}(\phi_H \mid \theta)}{\mathrm{Var}(\phi_H)} - \gamma$ governs feasibility: $\delta$ must outweigh the worst-case shared-error term $\gamma$ after scaling by the human-uncertainty factor $\mathrm{Var}(\phi_H \mid \theta)/\mathrm{Var}(\phi_H)$. When $\kappa_\gamma$ is small, the lower bound of $b_{\mathrm{pos}}$ collapses toward 0, and the symmetric policy cannot have enough uniformly safe evidence to overrule $d_H$.

# B. Proofs

### B.1. Proof of Proposition 3.3

Fix a covariance matrix $\Sigma$ for $(\theta, \phi_H, \phi_{AI})$ and write $X := (\phi_H, \phi_{AI})^\top$. A (deterministic) decision rule is a measurable function $d : \mathbb{R}^2 \to \mathbb{R}$, and its mean-squared prediction loss under $\Sigma$ is

$$L_\Sigma(d) = \mathbb{E}_\Sigma\big[(\theta - d(X))^2\big].$$

Note that by Assumption 3.1, the marginal law of $X = (\phi_H, \phi_{AI})$ is the same for all $\Sigma \in \mathcal{U}$. Denote this common marginal distribution by $P_X$ and write $\langle f, g \rangle := \mathbb{E}_{P_X}[f(X)g(X)]$ for the $L^2(P_X)$ inner product. Let $S := \mathrm{span}\{\phi_H, \phi_{AI}\} \subset L^2(P_X)$ be the two-dimensional linear subspace of (square-integrable) linear functions of $X$. Given any decision rule $d \in L^2(P_X)$, let $d^{\mathrm{lin}} \in S$ be its orthogonal projection onto $S$; equivalently, $d^{\mathrm{lin}}$ is the unique linear function $a_d\phi_H + b_d\phi_{AI}$ that minimizes $\mathbb{E}_{P_X}[(d(X) - a\phi_H - b\phi_{AI})^2]$. Define the residual $u := d - d^{\mathrm{lin}}$. By the characterization of orthogonal projections in Hilbert spaces, $u$ is orthogonal to $S$, i.e.,

$$\mathbb{E}_{P_X}[u(X)\phi_H] = 0 \qquad \text{and} \qquad \mathbb{E}_{P_X}[u(X)\phi_{AI}] = 0. \tag{10}$$

Fix any $\Sigma \in \mathcal{U}$. Since $(\theta, X)$ is jointly Gaussian under $\Sigma$, the conditional mean $\mu_\Sigma(X) := \mathbb{E}_\Sigma[\theta \mid X]$ is an element of $S$; that is, $\mu_\Sigma(X) = a_\Sigma\phi_H + b_\Sigma\phi_{AI}$ for some coefficients depending on $\Sigma$. The tower property gives

$$\mathbb{E}_\Sigma[u(X)\theta] = \mathbb{E}_\Sigma\big[u(X)\,\mathbb{E}_\Sigma[\theta \mid X]\big] = \mathbb{E}_{P_X}[u(X)\mu_\Sigma(X)].$$

But $\mu_\Sigma \in S$ and $u \perp S$ by (10), hence $\mathbb{E}_{P_X}[u\mu_\Sigma] = 0$ and therefore $\mathbb{E}_\Sigma[u\theta] = 0$. Similarly, since $d^{\mathrm{lin}} \in S$, we have $\mathbb{E}_{P_X}[ud^{\mathrm{lin}}] = 0$ and thus $\mathbb{E}_\Sigma[ud^{\mathrm{lin}}] = 0$. Expanding the squared loss and using these orthogonality relations yields

$$\begin{aligned}
L_\Sigma(d) &= \mathbb{E}_\Sigma\big[(\theta - d^{\mathrm{lin}}(X) - u(X))^2\big] \\
&= \mathbb{E}_\Sigma\big[(\theta - d^{\mathrm{lin}}(X))^2\big] + \mathbb{E}_{P_X}[u(X)^2] - 2\mathbb{E}_\Sigma\big[u(X)\theta\big] + 2\mathbb{E}_\Sigma\big[u(X)d^{\mathrm{lin}}(X)\big] \\
&= L_\Sigma(d^{\mathrm{lin}}) + \mathbb{E}_{P_X}[u(X)^2] \\
&\geq L_\Sigma(d^{\mathrm{lin}}).
\end{aligned}$$

Thus, for *every* $\Sigma \in \mathcal{U}$, the linear projection $d^{\mathrm{lin}}$ weakly improves on $d$.

### B.2. Auxiliary Lemmas for MSE Proofs

We record two elementary lemmas that will be used repeatedly in the proofs of Propositions 3.4–3.6. All expectations, variances, and covariances are taken with respect to the joint Gaussian law $N(0, \Sigma)$ under the covariance matrix currently under discussion.

**Lemma B.1** (Loss gap for linear aggregation). *Fix any covariance matrix $\Sigma$ for $(\theta, \phi_H, \phi_{AI})$. Let $d_H(\phi_H) := \phi_H$ and, for any $b \in \mathbb{R}$, define $d_b(\phi_H, \phi_{AI}) := \phi_H + b\phi_{AI}$. Under mean-squared error loss $\ell(d, \theta) = (d - \theta)^2$, the difference in expected losses satisfies*

$$L_\Sigma(d_b) - L_\Sigma(d_H) = b^2 Var(\phi_{AI}) - 2b\, Cov(\theta - \phi_H, \phi_{AI}). \tag{11}$$

*Proof.* Define the human prediction error $e_H := \theta - \phi_H$. Then for any real $b$ we have

$$\begin{aligned}
L_\Sigma(d_b) &= \mathbb{E}\big[(\theta - d_b(\phi_H, \phi_{AI}))^2\big] = \mathbb{E}\big[(\theta - \phi_H - b\phi_{AI})^2\big] \\
&= \mathbb{E}\big[(e_H - b\phi_{AI})^2\big].
\end{aligned}$$

We now expand the square *deterministically* and then take expectations:

$$(e_H - b\phi_{AI})^2 = e_H^2 + b^2\phi_{AI}^2 - 2b\, e_H\phi_{AI}.$$

Taking expectations and using $\mathbb{E}[\phi_{AI}^2] = Var(\phi_{AI})$ and $\mathbb{E}[e_H\phi_{AI}] = Cov(e_H, \phi_{AI})$ (all variables are mean-zero) yields

$$L_\Sigma(d_b) = \mathbb{E}[e_H^2] + b^2 Var(\phi_{AI}) - 2b\, Cov(e_H, \phi_{AI}).$$

On the other hand,

$$L_\Sigma(d_H) = \mathbb{E}\big[(\theta - \phi_H)^2\big] = \mathbb{E}[e_H^2].$$

Subtracting $L_\Sigma(d_H)$ from $L_\Sigma(d_b)$ and recalling $e_H = \theta - \phi_H$ gives exactly (11).

**Lemma B.2.** *Assume $(\theta, \phi_H, \phi_{AI}) \sim N(0, \Sigma)$. Then the covariance between the machine signal and the* human prediction error *satisfies the identity*

$$Cov(\theta - \phi_H, \phi_{AI}) = Cov(\theta, \phi_{AI}) \frac{Var(\phi_H \mid \theta)}{Var(\phi_H)} - Cov(\phi_H, \phi_{AI} \mid \theta). \tag{12}$$

*In particular, if $Cov(\phi_H, \phi_{AI} \mid \theta) \leq \gamma$ then*

$$Cov(\theta - \phi_H, \phi_{AI}) \geq Cov(\theta, \phi_{AI}) \frac{Var(\phi_H \mid \theta)}{Var(\phi_H)} - \gamma, \tag{13}$$

*and if additionally $Cov(\theta, \phi_{AI}) \geq \delta$ then*

$$Cov(\theta - \phi_H, \phi_{AI}) \geq \delta \frac{Var(\phi_H \mid \theta)}{Var(\phi_H)} - \gamma. \tag{14}$$

*Proof.*

**Step 1: decompose $\mathbf{Cov}(\phi_H, \phi_{AI})$ using the law of total covariance.** Since $\theta$ is one-dimensional, the law of total covariance gives

$$\mathrm{Cov}(\phi_H, \phi_{AI}) = \mathbb{E}\big[\mathrm{Cov}(\phi_H, \phi_{AI} \mid \theta)\big] + \mathrm{Cov}\big(\mathbb{E}[\phi_H \mid \theta], \mathbb{E}[\phi_{AI} \mid \theta]\big). \tag{15}$$

Under joint Gaussianity, the conditional covariance $\mathrm{Cov}(\phi_H, \phi_{AI} \mid \theta)$ is almost surely constant as a function of $\theta$, so the first term reduces to $\mathrm{Cov}(\phi_H, \phi_{AI} \mid \theta)$ itself. Moreover, again by joint Gaussianity (and mean-zero),

$$\mathbb{E}[\phi_H \mid \theta] = \frac{\mathrm{Cov}(\phi_H, \theta)}{\mathrm{Var}(\theta)} \theta, \qquad \mathbb{E}[\phi_{AI} \mid \theta] = \frac{\mathrm{Cov}(\phi_{AI}, \theta)}{\mathrm{Var}(\theta)} \theta.$$

Substituting these linear regressions into the second term of (15) yields

$$\begin{aligned} \mathrm{Cov}\big(\mathbb{E}[\phi_H \mid \theta], \mathbb{E}[\phi_{AI} \mid \theta]\big) &= \mathrm{Cov}\Big(\frac{\mathrm{Cov}(\phi_H, \theta)}{\mathrm{Var}(\theta)} \theta, \frac{\mathrm{Cov}(\phi_{AI}, \theta)}{\mathrm{Var}(\theta)} \theta\Big) \\ &= \frac{\mathrm{Cov}(\phi_H, \theta)\mathrm{Cov}(\phi_{AI}, \theta)}{\mathrm{Var}(\theta)^2} \mathrm{Var}(\theta) = \frac{\mathrm{Cov}(\phi_H, \theta)\mathrm{Cov}(\phi_{AI}, \theta)}{\mathrm{Var}(\theta)}. \end{aligned} \tag{16}$$

Combining (15) and (16) gives the identity

$$\mathrm{Cov}(\phi_H, \phi_{AI}) = \mathrm{Cov}(\phi_H, \phi_{AI} \mid \theta) + \frac{\mathrm{Cov}(\phi_H, \theta)\mathrm{Cov}(\phi_{AI}, \theta)}{\mathrm{Var}(\theta)}. \tag{17}$$

**Step 2: rewrite $\mathbf{Cov}(\theta - \phi_H, \phi_{AI})$ in terms of conditional covariance.** By bilinearity of covariance,

$$\mathrm{Cov}(\theta - \phi_H, \phi_{AI}) = \mathrm{Cov}(\theta, \phi_{AI}) - \mathrm{Cov}(\phi_H, \phi_{AI}). \tag{18}$$

Substituting (17) into (18) and using symmetry $\mathrm{Cov}(\theta, \phi_{AI}) = \mathrm{Cov}(\phi_{AI}, \theta)$ yields

$$\begin{aligned} \mathrm{Cov}(\theta - \phi_H, \phi_{AI}) &= \mathrm{Cov}(\theta, \phi_{AI}) - \mathrm{Cov}(\phi_H, \phi_{AI} \mid \theta) - \frac{\mathrm{Cov}(\phi_H, \theta)\mathrm{Cov}(\theta, \phi_{AI})}{\mathrm{Var}(\theta)} \\ &= \mathrm{Cov}(\theta, \phi_{AI})\Big(1 - \frac{\mathrm{Cov}(\phi_H, \theta)}{\mathrm{Var}(\theta)}\Big) - \mathrm{Cov}(\phi_H, \phi_{AI} \mid \theta). \end{aligned} \tag{19}$$

**Step 3: use normalization to identify the multiplicative factor.** By our normalization choice, $\mathrm{Cov}(\phi_H, \theta) = \mathrm{Var}(\phi_H)$. For a jointly Gaussian pair $(\phi_H, \theta)$, the conditional variance is

$$\mathrm{Var}(\phi_H \mid \theta) = \mathrm{Var}(\phi_H) - \frac{\mathrm{Cov}(\phi_H, \theta)^2}{\mathrm{Var}(\theta)}. \tag{20}$$

Substituting $\text{Cov}(\phi_H, \theta) = \text{Var}(\phi_H)$ into (20) gives

$$\text{Var}(\phi_H \mid \theta) = \text{Var}(\phi_H) - \frac{\text{Var}(\phi_H)^2}{\text{Var}(\theta)} = \text{Var}(\phi_H)\Big(1 - \frac{\text{Var}(\phi_H)}{\text{Var}(\theta)}\Big). \tag{21}$$

Dividing (21) by $\text{Var}(\phi_H) > 0$ yields

$$1 - \frac{\text{Cov}(\phi_H, \theta)}{\text{Var}(\theta)} = 1 - \frac{\text{Var}(\phi_H)}{\text{Var}(\theta)} = \frac{\text{Var}(\phi_H \mid \theta)}{\text{Var}(\phi_H)}. \tag{22}$$

Substituting (22) into (19) proves (12).

**Step 4: inequalities.** If $\text{Cov}(\phi_H, \phi_{AI} \mid \theta) \le \gamma$, then (12) implies

$$\text{Cov}(\theta - \phi_H, \phi_{AI}) \ge \text{Cov}(\theta, \phi_{AI})\frac{\text{Var}(\phi_H \mid \theta)}{\text{Var}(\phi_H)} - \gamma,$$

which is (13). If in addition $\text{Cov}(\theta, \phi_{AI}) \ge \delta$, then replacing $\text{Cov}(\theta, \phi_{AI})$ by $\delta$ in the right-hand side yields (14).

### B.3. Proof of Proposition 3.4

Fix an uncertainty set $\mathcal{U}$ as in the main text: all $\Sigma \in \mathcal{U}$ agree on every entry of the covariance matrix of $(\theta, \phi_H, \phi_{AI})$ except possibly on the single entry $\text{Cov}(\theta, \phi_{AI})$. In particular, $\text{Var}(\phi_{AI})$ is the same for all $\Sigma \in \mathcal{U}$.

Recall that the human-only decision rule is $d_H(\phi_H) = \phi_H$ (this uses our normalization $\text{Cov}(\theta, \phi_H)/\text{Var}(\phi_H) = 1$, so that $\mathbb{E}[\theta \mid \phi_H] = \phi_H$ under joint normality). For any $b \in \mathbb{R}$ define

$$d_b(\phi_H, \phi_{AI}) := \phi_H + b\phi_{AI}.$$

We will exhibit a nonzero coefficient $b$ such that $d_b$ strictly dominates $d_H$ over $\mathcal{U}$. First, note that by Lemma B.2,

$$\text{Cov}(\theta - \phi_H, \phi_{AI}) = \text{Cov}(\theta, \phi_{AI})\frac{\text{Var}(\phi_H \mid \theta)}{\text{Var}(\phi_H)} - \text{Cov}(\phi_H, \phi_{AI} \mid \theta).$$

Condition (2) states that $\text{Cov}(\phi_H, \phi_{AI} \mid \theta) \le 0$, hence for every $\Sigma \in \mathcal{U}$,

$$\text{Cov}(\theta - \phi_H, \phi_{AI}) \ge \text{Cov}(\theta, \phi_{AI})\frac{\text{Var}(\phi_H \mid \theta)}{\text{Var}(\phi_H)}. \tag{23}$$

Condition (1) gives $\text{Cov}(\theta, \phi_{AI}) \ge \delta$, so combining with (23) yields the uniform bound

$$\text{Cov}(\theta - \phi_H, \phi_{AI}) \ge \kappa \qquad \text{for all } \Sigma \in \mathcal{U}, \quad \text{where } \kappa := \delta\frac{\text{Var}(\phi_H \mid \theta)}{\text{Var}(\phi_H)}. \tag{24}$$

Note that $\kappa > 0$ whenever $\delta > 0$ and $\text{Var}(\phi_H \mid \theta) > 0$. Since $\text{Var}(\phi_{AI})$ is common across $\Sigma \in \mathcal{U}$, choose the nonzero coefficient

$$b^\star := \frac{\kappa}{\text{Var}(\phi_{AI})}.$$

Accordingly, we have that for every $\Sigma \in \mathcal{U}$,

$$\begin{aligned}
L_\Sigma(d_{b^\star}) - L_\Sigma(d_H) &= (b^\star)^2\text{Var}(\phi_{AI}) - 2b^\star\text{Cov}(\theta - \phi_H, \phi_{AI}) \quad \text{(Lemma B.1)}\\
&\le (b^\star)^2\text{Var}(\phi_{AI}) - 2b^\star\kappa\\
&= \frac{\kappa^2}{\text{Var}(\phi_{AI})} - 2\frac{\kappa^2}{\text{Var}(\phi_{AI})} = -\frac{\kappa^2}{\text{Var}(\phi_{AI})} < 0.
\end{aligned}$$

Hence $L_\Sigma(d_{b^\star}) < L_\Sigma(d_H)$ for all $\Sigma \in \mathcal{U}$, so $d_{b^\star}$ strictly dominates $d_H$ over $\mathcal{U}$.

## B.4. Proof of Proposition 3.6

Fix an uncertainty set $\mathcal{U}$ as in the statement of Proposition 3.6. In particular, all entries of $\Sigma$ are fixed across $\mathcal{U}$ except

$$s(\Sigma) := \mathrm{Cov}_\Sigma(\theta, \phi_{AI}).$$

Define

$$\delta := \inf_{\Sigma \in \mathcal{U}} s(\Sigma) \qquad \text{and} \qquad \gamma := \sup_{\Sigma \in \mathcal{U}} \mathrm{Cov}_\Sigma(\phi_H, \phi_{AI} \mid \theta).$$

Recall that $d_H(\phi_H) = \phi_H$ and that for any $b \in \mathbb{R}$ we write

$$d_b(\phi_H, \phi_{AI}) := \phi_H + b\phi_{AI}.$$

We prove both directions.

**Sufficiency.**  Assume first that $\gamma < \delta \, \frac{\mathrm{Var}(\phi_H \mid \theta)}{\mathrm{Var}(\phi_H)}$. We will show that every $b$ in the stated interval satisfies $L_\Sigma(d_b) < L_\Sigma(d_H)$ uniformly over $\Sigma \in \mathcal{U}$.

**Necessity.**  Conversely, assume that $\gamma \geq \delta \, \frac{\mathrm{Var}(\phi_H \mid \theta)}{\mathrm{Var}(\phi_H)}$ and fix an arbitrary coefficient $b > 0$. We will use only the one-free-entry structure of $\mathcal{U}$ (together with the fact that conditional covariances are affine in $s(\Sigma)$) to select some $\Sigma \in \mathcal{U}$ for which $L_\Sigma(d_b) > L_\Sigma(d_H)$.

**Step 0: conditional covariance is affine decreasing in $s(\Sigma)$.**  Since all entries of $\Sigma$ except $s(\Sigma) = \mathrm{Cov}_\Sigma(\theta, \phi_{AI})$ are fixed across $\mathcal{U}$, the (unconditional) covariance

$$c := \mathrm{Cov}(\phi_H, \phi_{AI})$$

is constant over $\Sigma \in \mathcal{U}$. By the covariance decomposition (17) established in the proof of Lemma B.2,

$$\mathrm{Cov}_\Sigma(\phi_H, \phi_{AI}) = \mathrm{Cov}_\Sigma(\phi_H, \phi_{AI} \mid \theta) + \frac{\mathrm{Cov}(\phi_H, \theta)\mathrm{Cov}_\Sigma(\phi_{AI}, \theta)}{\mathrm{Var}(\theta)}.$$

Rearranging and using symmetry $\mathrm{Cov}_\Sigma(\phi_{AI}, \theta) = \mathrm{Cov}_\Sigma(\theta, \phi_{AI}) = s(\Sigma)$ gives

$$\mathrm{Cov}_\Sigma(\phi_H, \phi_{AI} \mid \theta) = c - \frac{\mathrm{Cov}(\phi_H, \theta)}{\mathrm{Var}(\theta)} s(\Sigma). \tag{25}$$

Since under our normalization, $\mathrm{Cov}(\phi_H, \theta) = \mathrm{Var}(\phi_H)$, so if we define the (fixed) constant

$$\kappa := \frac{\mathrm{Var}(\phi_H)}{\mathrm{Var}(\theta)} > 0,$$

then (25) becomes

$$\mathrm{Cov}_\Sigma(\phi_H, \phi_{AI} \mid \theta) = c - \kappa\, s(\Sigma). \tag{26}$$

In particular, $\mathrm{Cov}_\Sigma(\phi_H, \phi_{AI} \mid \theta)$ is an affine (indeed, strictly) decreasing function of $s(\Sigma)$.

**Step 0a: the maximizer of $\mathrm{Cov}(\phi_H, \phi_{AI} \mid \theta)$ corresponds to the minimizer of $s(\Sigma)$.**  Since $s(\Sigma) \geq \delta$ for all $\Sigma \in \mathcal{U}$ and (26) is decreasing in $s(\Sigma)$, we have for every $\Sigma \in \mathcal{U}$ that

$$\mathrm{Cov}_\Sigma(\phi_H, \phi_{AI} \mid \theta) \leq c - \kappa\, \delta.$$

Taking the supremum over $\Sigma \in \mathcal{U}$ yields

$$\gamma \leq c - \kappa\, \delta. \tag{27}$$

For the reverse inequality, fix any $\varepsilon > 0$. By definition of $\delta$ as an infimum, there exists some $\Sigma_\varepsilon \in \mathcal{U}$ such that

$$s(\Sigma_\varepsilon) < \delta + \varepsilon. \tag{28}$$

Applying (26) at $\Sigma = \Sigma_\varepsilon$ and using (28) gives

$$\mathrm{Cov}_{\Sigma_\varepsilon}(\phi_H, \phi_{AI} \mid \theta) = c - \kappa\, s(\Sigma_\varepsilon) > c - \kappa(\delta + \varepsilon) = (c - \kappa\delta) - \kappa\varepsilon.$$

Taking the supremum over $\Sigma \in \mathcal{U}$ shows that $\gamma \geq (c - \kappa\delta) - \kappa\varepsilon$ for every $\varepsilon > 0$. Letting $\varepsilon \downarrow 0$ yields

$$\gamma \geq c - \kappa\, \delta.$$

Combining this with (27) yields the identity

$$\gamma = c - \kappa\, \delta. \tag{29}$$

Moreover, the construction above shows that for each $\varepsilon > 0$ we can choose $\Sigma_\varepsilon \in \mathcal{U}$ such that (28) holds and, simultaneously,

$$\mathrm{Cov}_{\Sigma_\varepsilon}(\phi_H, \phi_{AI} \mid \theta) > \gamma - \kappa\varepsilon. \tag{30}$$

**Step 1: an exact loss gap formula.** Fix $\Sigma \in \mathcal{U}$. Lemma B.1 gives

$$L_\Sigma(d_b) - L_\Sigma(d_H) = b^2 \mathrm{Var}(\phi_{AI}) - 2b\, \mathrm{Cov}(\theta - \phi_H, \phi_{AI}). \tag{31}$$

**Step 2: a uniform lower bound on $\mathrm{Cov}(\theta - \phi_H, \phi_{AI})$.** Lemma B.2 implies that, for every $\Sigma \in \mathcal{U}$,

$$\mathrm{Cov}(\theta - \phi_H, \phi_{AI}) = \mathrm{Cov}(\theta, \phi_{AI}) \frac{\mathrm{Var}(\phi_H \mid \theta)}{\mathrm{Var}(\phi_H)} - \mathrm{Cov}(\phi_H, \phi_{AI} \mid \theta).$$

By definition of $\delta$ and $\gamma$ we have, for every $\Sigma \in \mathcal{U}$,

$$\mathrm{Cov}(\theta, \phi_{AI}) = s(\Sigma) \geq \delta \qquad \text{and} \qquad \mathrm{Cov}(\phi_H, \phi_{AI} \mid \theta) \leq \gamma.$$

Substituting these bounds into the preceding identity yields the *uniform* inequality

$$\mathrm{Cov}(\theta - \phi_H, \phi_{AI}) \geq \kappa_\gamma \qquad \text{for all } \Sigma \in \mathcal{U}, \quad \text{where} \quad \kappa_\gamma := \delta\, \frac{\mathrm{Var}(\phi_H \mid \theta)}{\mathrm{Var}(\phi_H)} - \gamma. \tag{32}$$

Under the hypothesis of the sufficiency direction, $\kappa_\gamma > 0$.

**Step 3: choosing $b$ and verifying strict dominance (sufficiency).** Assume $\kappa_\gamma > 0$. Fix any coefficient $b$ satisfying

$$0 < b < \frac{2\kappa_\gamma}{\mathrm{Var}(\phi_{AI})}.$$

Since $\mathrm{Var}(\phi_{AI}) > 0$, combining (31) with (32) yields, for every $\Sigma \in \mathcal{U}$,

$$\begin{aligned}
L_\Sigma(d_b) - L_\Sigma(d_H) &= b^2 \mathrm{Var}(\phi_{AI}) - 2b\, \mathrm{Cov}(\theta - \phi_H, \phi_{AI}) \\
&\leq b^2 \mathrm{Var}(\phi_{AI}) - 2b\, \kappa_\gamma \\
&= b\, \mathrm{Var}(\phi_{AI}) \Big(b - \frac{2\kappa_\gamma}{\mathrm{Var}(\phi_{AI})}\Big) < 0,
\end{aligned}$$

where the final strict inequality uses $b > 0$ and $b < 2\kappa_\gamma/\mathrm{Var}(\phi_{AI})$. Thus $L_\Sigma(d_b) < L_\Sigma(d_H)$ for all $\Sigma \in \mathcal{U}$. This establishes the sufficiency direction (and the "moreover" part) of the proposition.

**Step 4: necessity of the threshold (the "only if" direction for $b > 0$).** Assume now that $\gamma \geq \delta\, \frac{\mathrm{Var}(\phi_H \mid \theta)}{\mathrm{Var}(\phi_H)}$, equivalently $\kappa_\gamma \leq 0$. Fix any coefficient $b > 0$.

**Step 4a: selecting an adversarial $\Sigma \in \mathcal{U}$.** Choose $\varepsilon > 0$ small enough that

$$\kappa_\gamma + \varepsilon < \frac{b}{2} \mathrm{Var}(\phi_{AI}). \tag{33}$$

Such an $\varepsilon$ exists because $\kappa_\gamma \leq 0$ while the right-hand side is strictly positive. Let $\Sigma_\varepsilon \in \mathcal{U}$ be the covariance matrix guaranteed by Step 0a, i.e., satisfying both (28) and (30). Write $\Sigma := \Sigma_\varepsilon$ for brevity.

**Step 4b: bounding $\mathrm{Cov}_\Sigma(\theta - \phi_H, \phi_{AI})$ from above.** Applying Lemma B.2 at $\Sigma$ gives

$$\mathrm{Cov}_\Sigma(\theta - \phi_H, \phi_{AI}) = s(\Sigma)\frac{\mathrm{Var}(\phi_H \mid \theta)}{\mathrm{Var}(\phi_H)} - \mathrm{Cov}_\Sigma(\phi_H, \phi_{AI} \mid \theta).$$

Using $s(\Sigma) < \delta + \varepsilon$ from (28) and $\mathrm{Cov}_\Sigma(\phi_H, \phi_{AI} \mid \theta) > \gamma - \kappa\varepsilon$ from (30) yields

$$\mathrm{Cov}_\Sigma(\theta - \phi_H, \phi_{AI}) < (\delta + \varepsilon)\frac{\mathrm{Var}(\phi_H \mid \theta)}{\mathrm{Var}(\phi_H)} - (\gamma - \kappa\varepsilon)$$

$$= \Big(\delta \frac{\mathrm{Var}(\phi_H \mid \theta)}{\mathrm{Var}(\phi_H)} - \gamma\Big) + \varepsilon\Big(\frac{\mathrm{Var}(\phi_H \mid \theta)}{\mathrm{Var}(\phi_H)} + \kappa\Big)$$

$$= \kappa_\gamma + \varepsilon\Big(\frac{\mathrm{Var}(\phi_H \mid \theta)}{\mathrm{Var}(\phi_H)} + \kappa\Big).$$

We now simplify the coefficient of $\varepsilon$. By (22) in the proof of Lemma B.2,

$$\frac{\mathrm{Var}(\phi_H \mid \theta)}{\mathrm{Var}(\phi_H)} = 1 - \frac{\mathrm{Var}(\phi_H)}{\mathrm{Var}(\theta)} = 1 - \kappa,$$

hence $\frac{\mathrm{Var}(\phi_H \mid \theta)}{\mathrm{Var}(\phi_H)} + \kappa = 1$. Therefore,

$$\mathrm{Cov}_\Sigma(\theta - \phi_H, \phi_{AI}) < \kappa_\gamma + \varepsilon. \tag{34}$$

Combining (34) with (33) yields

$$\mathrm{Cov}_\Sigma(\theta - \phi_H, \phi_{AI}) < \frac{b}{2}\mathrm{Var}(\phi_{AI}). \tag{35}$$

**Step 4c: concluding that $d_b$ cannot uniformly dominate $d_H$.** Finally, combining (31) with (35) gives

$$L_\Sigma(d_b) - L_\Sigma(d_H) = b^2\mathrm{Var}(\phi_{AI}) - 2b\,\mathrm{Cov}_\Sigma(\theta - \phi_H, \phi_{AI})$$

$$> b^2\mathrm{Var}(\phi_{AI}) - 2b \cdot \frac{b}{2}\mathrm{Var}(\phi_{AI}) = 0.$$

Thus, for this (adversarial) covariance matrix $\Sigma \in \mathcal{U}$, the decision rule $d_b$ performs strictly *worse* than $d_H$ under mean-squared error. Since $b > 0$ was arbitrary, it follows that when $\gamma \geq \delta \frac{\mathrm{Var}(\phi_H \mid \theta)}{\mathrm{Var}(\phi_H)}$ there is *no* positive coefficient $b$ for which $d_b$ strictly dominates $d_H$ over $\mathcal{U}$. This establishes the necessity direction.

**Deriving the maximum improvement in loss.** Next, we move on to discussing the maximum improvement in loss under the setting of positive correlation. Lemma B.1 (Eq. (11)) gives us that for every $\Sigma \in \mathcal{U}$,

$$L_\Sigma(d_H) - L_\Sigma(d_b) = 2b\,\mathrm{Cov}(\theta - \phi_H, \phi_{AI}) - b^2\mathrm{Var}(\phi_{AI}).$$

Using Lemma B.2 (Eq. (14)), we have the lower bound

$$\mathrm{Cov}(\theta - \phi_H, \phi_{AI}) \;\geq\; \delta\,\frac{\mathrm{Var}(\phi_H \mid \theta)}{\mathrm{Var}(\phi_H)} - \gamma \qquad \text{for all } \Sigma \in \mathcal{U}.$$

Substituting in this bound results in

$$\inf_{\Sigma \in \mathcal{U}} \Big\{L_\Sigma(d_H) - L_\Sigma(d_b)\Big\} \;\geq\; 2b\Big(\delta\,\frac{\mathrm{Var}(\phi_H \mid \theta)}{\mathrm{Var}(\phi_H)} - \gamma\Big) \;-\; b^2\,\mathrm{Var}(\phi_{AI}). \tag{36}$$

Next, we can note that the choice of

$$b^\star \triangleq \frac{\delta\,\frac{\mathrm{Var}(\phi_H \mid \theta)}{\mathrm{Var}(\phi_H)} - \gamma}{\mathrm{Var}(\phi_{AI})} \tag{37}$$

lies in the admissible interval from Proposition 3.6. We can note that right-hand side of (36) is a concave quadratic in $b$, which is maximized at $b = b^\star$ in (37). Thus, using this value guarantees that

$$L_\Sigma(d_{b^\star}) \;\leq\; L_\Sigma(d_H) \;-\; \frac{\left(\delta\, \frac{\mathrm{Var}(\phi_H\mid\theta)}{\mathrm{Var}(\phi_H)} - \gamma\right)^2}{\mathrm{Var}(\phi_{AI})} \qquad \forall \Sigma \in \mathcal{U}. \tag{38}$$

With $(\delta, \gamma, \mathrm{Var}(\phi_H), \mathrm{Var}(\phi_{AI}))$ fixed, the expression in (38) is the square of an affine function of $\mathrm{Var}(\phi_H \mid \theta)$ with positive slope $\delta/\mathrm{Var}(\phi_H)$. Hence it is strictly increasing in $\mathrm{Var}(\phi_H \mid \theta)$ on the region where the strict inequality in Proposition 3.6 holds.

### B.5. Proof of Proposition 3.7

Fix a covariance matrix $\Sigma$. By the definition of $r$ and the normalization $\mathrm{Cov}(\theta, \phi_H) = \mathrm{Var}(\phi_H)$,

$$\mathrm{Cov}_\Sigma(r, \theta) = \mathrm{Cov}_\Sigma(\phi_{AI}, \theta) - \frac{\mathrm{Cov}(\phi_H, \phi_{AI})}{\mathrm{Var}(\phi_H)} \mathrm{Cov}(\phi_H, \theta)$$

$$= \mathrm{Cov}_\Sigma(\phi_{AI}, \theta) - \mathrm{Cov}(\phi_H, \phi_{AI}) = \mathrm{Cov}_\Sigma(\theta - \phi_H, \phi_{AI}).$$

Thus $\beta(\Sigma) \neq 0$ implies $\mathrm{Cov}_\Sigma(\theta - \phi_H, \phi_{AI}) \neq 0$ whenever $\mathrm{Var}(r) \neq 0$. We assume this non-redundancy condition; otherwise, the AI signal contains no linear component beyond $\phi_H$. Lemma B.1 gives, for any $b \in \mathbb{R}$,

$$L_\Sigma(d_b) - L_\Sigma(d_H) = b^2 \mathrm{Var}(\phi_{AI}) - 2b\, \mathrm{Cov}_\Sigma(\theta - \phi_H, \phi_{AI}).$$

This is a strictly convex quadratic in $b$ with minimizer

$$b^\star \triangleq \frac{\mathrm{Cov}_\Sigma(\theta - \phi_H, \phi_{AI})}{\mathrm{Var}(\phi_{AI})},$$

Since $\mathrm{Cov}_\Sigma(\theta - \phi_H, \phi_{AI}) \neq 0$, we have $b^\star \neq 0$. Plugging $b^\star$ into the loss gap yields

$$L_\Sigma(d_{b^\star}) - L_\Sigma(d_H) = -\frac{\mathrm{Cov}_\Sigma(\theta - \phi_H, \phi_{AI})^2}{\mathrm{Var}(\phi_{AI})} < 0.$$

Therefore $d_{b^\star}$ strictly improves upon $d_H$ under $\Sigma$, proving the proposition.

### B.6. Proof of Proposition 3.8

We first provide introduce some additional notation and supporting lemmas. Using joint normality and the definition of the residual $r$, for any fixed covariance matrix $\Sigma$ we have

$$\theta \mid (\phi_H, r) \sim N\big(\mu_\Sigma(\phi_H, r),\ \sigma_\Sigma^2(\phi_H, r)\big), \tag{39}$$

$$\mu_\Sigma(\phi_H, r) = \phi_H + \beta(\Sigma)\, r, \qquad \sigma_\Sigma^2(\phi_H, r) = \sigma_H^2 - \beta(\Sigma)^2\, \mathrm{Var}(r).$$

so that for all $\Sigma \in \mathcal{U}$ and all $(\phi_H, r)$,

$$\mu_\Sigma(\phi_H, r) \begin{cases} \geq \mu_{\mathrm{b}}(\phi_H, r) & \text{if } r \geq 0, \\ \leq \mu_{\mathrm{b}}(\phi_H, r) & \text{if } r \leq 0, \end{cases} \qquad 0 < \sigma_\Sigma^2(\phi_H, r) \leq \sigma_b^2 < \sigma_H^2. \tag{40}$$

For any fixed covariance matrix $\Sigma$ and any pair $(x, y) \in \mathbb{R}^2$, write

$$P_\Sigma(x, y) \triangleq \mathbb{P}_\Sigma\big(\theta \geq \tau \mid \phi_H = x, r = y\big).$$

Using (39) we may express this as

$$P_\Sigma(x, y) = \Phi\left(\frac{\mu_\Sigma(x, y) - \tau}{\sigma_\Sigma(x, y)}\right).$$

We will bound $P_\Sigma(x, y)$ uniformly over $\Sigma \in \mathcal{U}$ using the pessimistic mean–variance bounds (40).

*Proof of Lemma 3.5.* Using (1) and $\text{Cov}(\theta, \phi_H) = \text{Var}(\phi_H)$,

$$\text{Cov}_\Sigma(r, \theta) = \text{Cov}_\Sigma(\phi_{AI}, \theta) - \frac{\text{Cov}(\phi_H, \phi_{AI})}{\text{Var}(\phi_H)} \text{Cov}(\phi_H, \theta) = \text{Cov}_\Sigma(\phi_{AI}, \theta) - \text{Cov}(\phi_H, \phi_{AI}).$$

Moreover, the Gaussian identity for conditional covariance gives

$$\text{Cov}_\Sigma(\phi_H, \phi_{AI} \mid \theta) = \text{Cov}(\phi_H, \phi_{AI}) - \frac{\text{Cov}(\phi_H, \theta)\text{Cov}_\Sigma(\phi_{AI}, \theta)}{\text{Var}(\theta)} = \text{Cov}(\phi_H, \phi_{AI}) - \frac{\text{Var}(\phi_H)\text{Cov}_\Sigma(\phi_{AI}, \theta)}{\text{Var}(\theta)}.$$

Thus the assumption $\text{Cov}_\Sigma(\phi_H, \phi_{AI} \mid \theta) \leq 0$ implies

$$\text{Cov}(\phi_H, \phi_{AI}) \leq \frac{\text{Var}(\phi_H)}{\text{Var}(\theta)} \text{Cov}_\Sigma(\phi_{AI}, \theta),$$

and therefore

$$\text{Cov}_\Sigma(r, \theta) \geq \text{Cov}_\Sigma(\phi_{AI}, \theta)\left(1 - \frac{\text{Var}(\phi_H)}{\text{Var}(\theta)}\right) = \text{Cov}_\Sigma(\phi_{AI}, \theta)\frac{\text{Var}(\phi_H \mid \theta)}{\text{Var}(\phi_H)} \geq \delta \frac{\text{Var}(\phi_H \mid \theta)}{\text{Var}(\phi_H)}.$$

Define the corresponding pessimistic lower bound on the regression coefficient,

$$b \triangleq \delta \frac{\text{Var}(\phi_H \mid \theta)}{\text{Var}(\phi_H)\text{Var}(r)} > 0, \tag{41}$$

so that $\beta(\Sigma) \geq b$ for all $\Sigma \in \mathcal{U}$ by (2). $\qquad\square$

**Lemma B.3.** *Fix $\tau \in \mathbb{R}$ and define $F(\mu, \sigma) \triangleq \Phi\left(\frac{\mu - \tau}{\sigma}\right)$ for $\sigma > 0$.*

1. *For every $\sigma > 0$, $F(\cdot, \sigma)$ is strictly increasing in $\mu$.*

2. *For every $\mu > \tau$, $F(\mu, \cdot)$ is strictly decreasing in $\sigma$.*

3. *For every $\mu < \tau$, $F(\mu, \cdot)$ is strictly increasing in $\sigma$.*

*Proof of Lemma B.3.* Write $z(\mu, \sigma) \triangleq (\mu - \tau)/\sigma$. Then

$$F(\mu, \sigma) = \Phi(z(\mu, \sigma)).$$

Since $\Phi'(z) = \varphi(z) > 0$ for all $z$, it suffices to examine $\partial z/\partial\mu$ and $\partial z/\partial\sigma$. A direct calculation gives

$$\frac{\partial z}{\partial \mu} = \frac{1}{\sigma} > 0,$$

which yields (1). Next,

$$\frac{\partial z}{\partial \sigma} = -\frac{\mu - \tau}{\sigma^2}.$$

If $\mu > \tau$, then $\mu - \tau > 0$, so $\partial z/\partial\sigma < 0$ and hence $\partial F/\partial\sigma = \varphi(z)\,\partial z/\partial\sigma < 0$, proving (2). If $\mu < \tau$, then $\mu - \tau < 0$, so $\partial z/\partial\sigma > 0$ and therefore $\partial F/\partial\sigma > 0$, proving (3). $\qquad\square$

**Lemma B.4** (Worst-case lower bound for positive residuals)**.** *For every $\Sigma \in \mathcal{U}$ and every $(x, y)$ with $y \geq 0$,*

$$P_\Sigma(x, y) \geq P_{\text{low}}(x, y).$$

*Proof of Lemma B.4.* Fix $(x, y)$ with $y \geq 0$ and $\Sigma \in \mathcal{U}$. From (40) we have

$$\mu_\Sigma(x, y) = x + \beta(\Sigma)y \geq x + by = \mu_{\text{b}}(x, y), \qquad 0 < \sigma_\Sigma(x, y) \leq \sigma_b.$$

If $\mu_{\mathrm{b}}(x,y) < \tau$, then $P_{\mathrm{low}}(x,y) = 0$ by definition and $P_\Sigma(x,y) \geq 0$ trivially, so the inequality holds. If instead $\mu_{\mathrm{b}}(x,y) \geq \tau$, then by Lemma B.3(1)–(2) the map $F(\mu,\sigma) = \Phi((\mu - \tau)/\sigma)$ is increasing in $\mu$ and decreasing in $\sigma$ on the domain $\{\mu \geq \tau, \sigma > 0\}$. Since $(\mu_\Sigma(x,y), \sigma_\Sigma(x,y))$ lies in this domain and satisfies

$$\mu_\Sigma(x,y) \geq \mu_{\mathrm{b}}(x,y), \qquad \sigma_\Sigma(x,y) \leq \sigma_b,$$

we obtain

$$P_\Sigma(x,y) = F(\mu_\Sigma(x,y), \sigma_\Sigma(x,y)) \geq F(\mu_{\mathrm{b}}(x,y), \sigma_b) = P_{\mathrm{low}}(x,y),$$

as claimed. □

**Lemma B.5** (Worst-case upper bound for negative residuals). *For every $\Sigma \in \mathcal{U}$ and every $(x,y)$ with $y < 0$,*

$$P_\Sigma(x,y) \leq P_{\mathrm{high}}(x,y).$$

*Proof of Lemma B.5.* Fix $(x,y)$ with $y < 0$ and $\Sigma \in \mathcal{U}$. Then $\beta(\Sigma) \geq b$ implies

$$\mu_\Sigma(x,y) = x + \beta(\Sigma)y \leq x + by = \mu_{\mathrm{b}}(x,y),$$

and we always have $0 < \sigma_\Sigma(x,y) \leq \sigma_H$ because conditioning on $(\phi_H, r)$ cannot increase the variance beyond $\mathrm{Var}(\theta \mid \phi_H) = \sigma_H^2$.

Consider first the case $\mu_{\mathrm{b}}(x,y) \geq \tau$. Then $P_{\mathrm{high}}(x,y) = 1$, and trivially $P_\Sigma(x,y) \leq 1 = P_{\mathrm{high}}(x,y)$.

Now suppose $\mu_{\mathrm{b}}(x,y) < \tau$. For any covariance structure consistent with our uncertainty set we can write

$$P_\Sigma(x,y) = F(\mu_\Sigma(x,y), \sigma_\Sigma(x,y)) \leq \sup_{\mu \leq \mu_{\mathrm{b}}(x,y),\, 0 < \sigma \leq \sigma_H} F(\mu, \sigma),$$

where again $F(\mu, \sigma) = \Phi((\mu - \tau)/\sigma)$. By Lemma B.3(1), $F$ is increasing in $\mu$ for fixed $\sigma$, so the supremum over $\mu \leq \mu_{\mathrm{b}}(x,y)$ is attained at $\mu = \mu_{\mathrm{b}}(x,y)$. Since now $\mu_{\mathrm{b}}(x,y) < \tau$, Lemma B.3(3) implies that $F(\mu_{\mathrm{b}}(x,y), \sigma)$ is increasing in $\sigma$, so the supremum over $\sigma \leq \sigma_H$ is obtained at $\sigma = \sigma_H$. Thus

$$\sup_{\mu \leq \mu_{\mathrm{b}}(x,y),\, 0 < \sigma \leq \sigma_H} F(\mu, \sigma) = F(\mu_{\mathrm{b}}(x,y), \sigma_H) = \Phi\left(\frac{\mu_{\mathrm{b}}(x,y) - \tau}{\sigma_H}\right) = P_{\mathrm{high}}(x,y).$$

Since $(\mu_\Sigma(x,y), \sigma_\Sigma(x,y))$ satisfies the constraints $\mu_\Sigma(x,y) \leq \mu_{\mathrm{b}}(x,y)$ and $0 < \sigma_\Sigma(x,y) \leq \sigma_H$, we conclude

$$P_\Sigma(x,y) \leq P_{\mathrm{high}}(x,y),$$

as claimed. □

*Proof of Proposition 3.8.* Fix $\Sigma \in \mathcal{U}$. Recall the utility from (3),

$$u(1,\theta) = 1\{\theta \geq \tau\} - c, \qquad u(0,\theta) = 0.$$

For any decision rule $d$ and any realization $(\phi_H, r) = (x,y)$, taking conditional expectations yields

$$\mathbb{E}_\Sigma[u(d,\theta) \mid \phi_H = x, r = y] = (P_\Sigma(x,y) - c)\, d(x,y),$$

where $P_\Sigma(x,y) = \mathbb{P}_\Sigma(\theta \geq \tau \mid \phi_H = x, r = y)$. Therefore,

$$\mathbb{E}_\Sigma\big[u(d_J(\phi_H, \phi_{AI}), \theta)\big] - \mathbb{E}_\Sigma\big[u(d_H(\phi_H), \theta)\big]$$
$$= \mathbb{E}_\Sigma\Big[(P_\Sigma(\phi_H, r) - c)\,(d_J(\phi_H, \phi_{AI}) - d_H(\phi_H))\Big].$$

Define the (deterministic) regions on which $d_J$ differs from $d_H$:

$$A_+ \triangleq \{d_J(\phi_H, \phi_{AI}) = 1,\ d_H(\phi_H) = 0\}, \qquad A_- \triangleq \{d_J(\phi_H, \phi_{AI}) = 0,\ d_H(\phi_H) = 1\}.$$

On the complement of $A_+ \cup A_-$, the two rules coincide and the integrand is zero. Hence

$$\mathbb{E}_\Sigma\big[u(d_J(\phi_H, \phi_{AI}), \theta)\big] - \mathbb{E}_\Sigma\big[u(d_H(\phi_H), \theta)\big]$$
$$= \mathbb{E}_\Sigma\big[(P_\Sigma(\phi_H, r) - c)\,\mathbb{1}\{(\phi_H, r) \in A_+\}\big] \ + \ \mathbb{E}_\Sigma\big[(c - P_\Sigma(\phi_H, r))\,\mathbb{1}\{(\phi_H, r) \in A_-\}\big]. \tag{42}$$

*Step 1: Added-action region $A_+$.* By definition (6), $A_+$ can occur only if $P_{\text{low}}(\phi_H, r) \geq c$. In particular, this implies $r \geq 0$, so Lemma B.4 applies and yields
$$P_\Sigma(\phi_H, r) \ \geq \ P_{\text{low}}(\phi_H, r) \ \geq \ c \quad \text{on } A_+.$$
Thus $P_\Sigma(\phi_H, r) - c \geq 0$ on $A_+$, and therefore the first term in (42) is nonnegative.

*Step 2: Removed-action region $A_-$.* By definition (6), $A_-$ can occur only if $d_H(\phi_H) = 1$, $r < 0$, and $P_{\text{high}}(\phi_H, r) \leq c$. Since $r < 0$, Lemma B.5 applies and gives
$$P_\Sigma(\phi_H, r) \ \leq \ P_{\text{high}}(\phi_H, r) \ \leq \ c \quad \text{on } A_-.$$
Hence $c - P_\Sigma(\phi_H, r) \geq 0$ on $A_-$, and the second term in (42) is also nonnegative.

Combining Steps 1–2 shows that the right-hand side of (42) is nonnegative for every $\Sigma \in \mathcal{U}$, proving weak dominance.

*Step 3: Strict improvement under $\beta(\Sigma_0) > b$.* Now suppose there exists $\Sigma_0 \in \mathcal{U}$ such that $\beta(\Sigma_0) > b$. Since $(\phi_H, r)$ is jointly normal with a nondegenerate covariance matrix and $r$ is independent of $\phi_H$, there exists $\varepsilon > 0$ such that
$$\mathbb{P}_{\Sigma_0}\big(\phi_H \in (x_H - \varepsilon, x_H)\big) > 0.$$

Moreover, because $r$ has unbounded support, we can choose $R > 0$ sufficiently large so that for every $x \in (x_H - \varepsilon, x_H)$ and every $y \geq R$ we have both $\mu_{\text{b}}(x, y) \geq \tau$ and $P_{\text{low}}(x, y) \geq c$.

Define the event
$$B \triangleq \{\phi_H \in (x_H - \varepsilon, x_H),\ r \geq R\}.$$

By independence of $\phi_H$ and $r$ and the positivity of both marginal probabilities, we have $\mathbb{P}_{\Sigma_0}(B) > 0$. On $B$ we have $d_H(\phi_H) = 0$ (since $\phi_H < x_H$) and $P_{\text{low}}(\phi_H, r) \geq c$, hence $d_J(\phi_H, \phi_{AI}) = 1$ by (6). That is, $B \subseteq A_+$.

Fix now any realization $(\phi_H, r) = (x, y) \in B$. By construction $y > 0$ and $\mu_{\text{b}}(x, y) \geq \tau$, and since $\beta(\Sigma_0) > b$ we have
$$\mu_{\Sigma_0}(x, y) = x + \beta(\Sigma_0)y \ > \ x + by = \mu_{\text{b}}(x, y), \qquad \sigma_{\Sigma_0}(x, y) \ < \ \sigma_b,$$

where the strict inequality for $\sigma_{\Sigma_0}(x, y)$ follows from $\sigma_{\Sigma_0}^2(x, y) = \sigma_H^2 - \beta(\Sigma_0)^2 \text{Var}(r)$ and $\beta(\Sigma_0)^2 > b^2$. By Lemma B.3(1)–(2), $F(\mu, \sigma) = \Phi((\mu - \tau)/\sigma)$ is strictly increasing in $\mu$ and strictly decreasing in $\sigma$ on $\{\mu > \tau, \sigma > 0\}$, so we obtain the strict inequality
$$P_{\Sigma_0}(x, y) = \Phi\left(\frac{\mu_{\Sigma_0}(x, y) - \tau}{\sigma_{\Sigma_0}(x, y)}\right) > \Phi\left(\frac{\mu_{\text{b}}(x, y) - \tau}{\sigma_b}\right) = P_{\text{low}}(x, y) \ \geq \ c.$$

In particular, $P_{\Sigma_0}(\phi_H, r) - c > 0$ on the event $B$, and since $B \subseteq A_+$ and $\mathbb{P}_{\Sigma_0}(B) > 0$, the first term in (42) is strictly positive under $\Sigma_0$. The second term is nonnegative by Step 2, and therefore
$$\mathbb{E}_{\Sigma_0}\big[u(d_J(\phi_H, \phi_{AI}), \theta)\big] \ > \ \mathbb{E}_{\Sigma_0}\big[u(d_H(\phi_H), \theta)\big].$$

This completes the proof. $\qquad\square$

## B.7. Proof of Proposition 3.9

We begin with a basic monotonicity lemma that holds for arbitrary distributions.

**Lemma B.6** (Monotone covariance). *Let $X$ be any real-valued random variable with $\mathbb{E}[X^2] < \infty$, and let $u, v : \mathbb{R} \to \mathbb{R}$ be nondecreasing functions such that $u(X)$ and $v(X)$ are square-integrable. Then*
$$\text{Cov}\big(u(X), v(X)\big) \ \geq \ 0,$$

*with strict inequality whenever $u$ and $v$ are strictly increasing on a set of positive probability under the law of $X$.*

*Proof.* Let $X'$ be an independent copy of $X$, defined on the same probability space. Then

$$\text{Cov}(u(X), v(X)) = \mathbb{E}\big[u(X)v(X)\big] - \mathbb{E}[u(X)]\,\mathbb{E}[v(X)]$$
$$= \frac{1}{2}\mathbb{E}\big[(u(X) - u(X'))(v(X) - v(X'))\big].$$

The equality in the last line is a standard polarization identity: expanding the right-hand side and using independence of $X$ and $X'$ yields exactly the covariance. Since $u$ and $v$ are nondecreasing, for every pair $(x, x')$ we have

$$(u(x) - u(x'))(v(x) - v(x')) \geq 0.$$

Thus the integrand in the last display is almost surely nonnegative, and hence its expectation is nonnegative:

$$\text{Cov}(u(X), v(X)) \geq 0.$$

If $u$ and $v$ are strictly increasing on a set $A$ with $\mathbb{P}(X \in A) > 0$, then there is a set of pairs $(x, x')$ of positive probability with $x \neq x'$ and $(u(x) - u(x'))(v(x) - v(x')) > 0$, implying strict inequality in the covariance. $\qquad\square$

We now apply Lemma B.6 conditionally to obtain a lower bound on the conditional covariance $\text{Cov}(\phi_{AI}, \theta \mid \phi_H)$.

**Lemma B.7.** *Under the assumptions of this section, for every value $x$ of $\phi_H$ for which $\text{Var}(\theta \mid \phi_H = x) < \infty$ we have*

$$\text{Cov}\big(\phi_{AI}, \theta \mid \phi_H = x\big) \geq k\,\text{Var}(\theta \mid \phi_H = x).$$

*Proof.* Fix $x$ and consider the conditional distribution of $\theta$ given $\phi_H = x$. Since $\phi_H = f(\theta) + \epsilon_H$ with $(\epsilon_H, \epsilon_{AI})$ independent of $\theta$, this conditional distribution is well-defined and has finite variance by assumption.

From the additive signal structure we have

$$\phi_{AI} = g(\theta) + \epsilon_{AI}.$$

Conditioning on $\phi_H = x$ yields

$$\text{Cov}(\phi_{AI}, \theta \mid \phi_H = x) = \text{Cov}\big(g(\theta) + \epsilon_{AI}, \theta \mid \phi_H = x\big)$$
$$= \text{Cov}\big(g(\theta), \theta \mid \phi_H = x\big) + \text{Cov}\big(\epsilon_{AI}, \theta \mid \phi_H = x\big).$$

We lower bound the second term using the assumption that $m(e) \triangleq \mathbb{E}[\epsilon_{AI} \mid \epsilon_H = e]$ is nonincreasing.

First observe that, given $(\theta, \phi_H = x)$, the value of $\epsilon_H$ is pinned down deterministically as

$$\epsilon_H = x - f(\theta).$$

Since $(\epsilon_H, \epsilon_{AI})$ is independent of $\theta$, the conditional distribution of $\epsilon_{AI}$ given $\epsilon_H$ does not depend on $\theta$, and therefore

$$\mathbb{E}[\epsilon_{AI} \mid \theta, \phi_H = x] = \mathbb{E}[\epsilon_{AI} \mid \epsilon_H = x - f(\theta)] = m\big(x - f(\theta)\big).$$

We now compute $\text{Cov}(\epsilon_{AI}, \theta \mid \phi_H = x)$ by conditioning on $\theta$. By the tower property,

$$\mathbb{E}[\epsilon_{AI}\,\theta \mid \phi_H = x] = \mathbb{E}\big[\mathbb{E}[\epsilon_{AI}\,\theta \mid \theta, \phi_H = x] \mid \phi_H = x\big]$$
$$= \mathbb{E}\big[\theta\,\mathbb{E}[\epsilon_{AI} \mid \theta, \phi_H = x] \mid \phi_H = x\big]$$
$$= \mathbb{E}\big[\theta\,m(x - f(\theta)) \mid \phi_H = x\big],$$

where the second equality uses that conditioning on $\theta$ makes $\theta$ constant. Similarly,

$$\mathbb{E}[\epsilon_{AI} \mid \phi_H = x] = \mathbb{E}\big[\mathbb{E}[\epsilon_{AI} \mid \theta, \phi_H = x] \mid \phi_H = x\big] = \mathbb{E}\big[m(x - f(\theta)) \mid \phi_H = x\big].$$

Combining the last two displays yields

$$\text{Cov}(\epsilon_{AI}, \theta \mid \phi_H = x) = \mathbb{E}[\epsilon_{AI}\,\theta \mid \phi_H = x] - \mathbb{E}[\epsilon_{AI} \mid \phi_H = x]\,\mathbb{E}[\theta \mid \phi_H = x]$$
$$= \mathbb{E}\big[\theta\,m(x - f(\theta)) \mid \phi_H = x\big] - \mathbb{E}\big[m(x - f(\theta)) \mid \phi_H = x\big]\mathbb{E}[\theta \mid \phi_H = x]$$
$$= \text{Cov}\big(m(x - f(\theta)), \theta \mid \phi_H = x\big).$$

Because $f$ is strictly increasing, the mapping $\theta \mapsto x - f(\theta)$ is nonincreasing. Because $m$ is nonincreasing by assumption, the composition $\theta \mapsto m(x - f(\theta))$ is therefore nondecreasing. Note that $m(\epsilon_H) = \mathbb{E}[\epsilon_{AI} \mid \epsilon_H]$ is square-integrable by Jensen's inequality, since $\mathbb{E}[\epsilon_{AI}^2] < \infty$. Consequently $m(x - f(\theta))$ is square-integrable under the conditional law of $\theta \mid \phi_H = x$. Applying Lemma B.6 conditionally (with $X = \theta \mid \phi_H = x$, $u(\theta) = m(x - f(\theta))$ and $v(\theta) = \theta$) yields

$$\mathrm{Cov}\big(m(x - f(\theta)), \theta \mid \phi_H = x\big) \geq 0,$$

and hence

$$\mathrm{Cov}(\epsilon_{AI}, \theta \mid \phi_H = x) \geq 0.$$

Combining the last two equations with the earlier decomposition gives

$$\mathrm{Cov}(\phi_{AI}, \theta \mid \phi_H = x) \geq \mathrm{Cov}\big(g(\theta), \theta \mid \phi_H = x\big).$$

Next, use the derivative lower bound (7) to decompose $g$ as

$$g(t) = kt + h(t),$$

where $h'(t) = g'(t) - k \geq 0$ for all $t$. Hence $h$ is nondecreasing. For the conditional law of $\theta \mid \phi_H = x$ we therefore have

$$\begin{aligned}
\mathrm{Cov}(g(\theta), \theta \mid \phi_H = x) &= \mathrm{Cov}\big(k\theta + h(\theta), \theta \mid \phi_H = x\big) \\
&= k\,\mathrm{Var}(\theta \mid \phi_H = x) + \mathrm{Cov}\big(h(\theta), \theta \mid \phi_H = x\big).
\end{aligned}$$

Applying Lemma B.6 conditionally (with $X = \theta \mid \phi_H = x$, $u = h$ and $v = \mathrm{id}$, both nondecreasing) shows that

$$\mathrm{Cov}\big(h(\theta), \theta \mid \phi_H = x\big) \geq 0.$$

Combining the last two displays yields

$$\mathrm{Cov}(g(\theta), \theta \mid \phi_H = x) \geq k\,\mathrm{Var}(\theta \mid \phi_H = x).$$

Finally, combining this inequality with $\mathrm{Cov}(\phi_{AI}, \theta \mid \phi_H = x) \geq \mathrm{Cov}(g(\theta), \theta \mid \phi_H = x)$ gives

$$\mathrm{Cov}\big(\phi_{AI}, \theta \mid \phi_H = x\big) \geq k\,\mathrm{Var}(\theta \mid \phi_H = x),$$

as claimed. $\square$

We now integrate these conditional covariances over the marginal distribution of $\phi_H$ to obtain a lower bound on $\mathrm{Cov}(\tilde{r}, \theta)$. First, observe that by definition,

$$\tilde{r} = \phi_{AI} - \mathbb{E}[\phi_{AI} \mid \phi_H],$$

so $\mathbb{E}[\tilde{r} \mid \phi_H] = 0$ and hence $\mathbb{E}[\tilde{r}] = 0$. Using the law of total covariance we can write

$$\begin{aligned}
\mathrm{Cov}(\tilde{r}, \theta) &= \mathbb{E}\big[\tilde{r}\,\theta\big] - \mathbb{E}[\tilde{r}]\,\mathbb{E}[\theta] \\
&= \mathbb{E}\big[\mathbb{E}[\tilde{r}\,\theta \mid \phi_H]\big] - 0 \cdot \mathbb{E}[\theta] \\
&= \mathbb{E}\big[\mathrm{Cov}(\tilde{r}, \theta \mid \phi_H)\big],
\end{aligned}$$

where in the last step we have used the fact that $\mathbb{E}[\tilde{r} \mid \phi_H] = 0$ implies

$$\mathbb{E}[\tilde{r}\,\theta \mid \phi_H] = \mathrm{Cov}(\tilde{r}, \theta \mid \phi_H) + \mathbb{E}[\tilde{r} \mid \phi_H]\,\mathbb{E}[\theta \mid \phi_H] = \mathrm{Cov}(\tilde{r}, \theta \mid \phi_H).$$

Next, note that for each fixed value $\phi_H = x$,

$$\begin{aligned}
\mathrm{Cov}(\tilde{r}, \theta \mid \phi_H = x) &= \mathrm{Cov}\big(\phi_{AI} - \mathbb{E}[\phi_{AI} \mid \phi_H = x], \theta \mid \phi_H = x\big) \\
&= \mathrm{Cov}(\phi_{AI}, \theta \mid \phi_H = x) - \mathrm{Cov}\big(\mathbb{E}[\phi_{AI} \mid \phi_H = x], \theta \mid \phi_H = x\big).
\end{aligned}$$

The second term vanishes because $\mathbb{E}[\phi_{AI} \mid \phi_H = x]$ is a constant (given $\phi_H = x$), so

$$\text{Cov}(\tilde{r}, \theta \mid \phi_H = x) = \text{Cov}(\phi_{AI}, \theta \mid \phi_H = x).$$

Applying Lemma B.7 to the right-hand side yields

$$\text{Cov}(\tilde{r}, \theta \mid \phi_H = x) \geq k \, \text{Var}(\theta \mid \phi_H = x),$$

for every $x$ with finite conditional variance. Taking expectations over $\phi_H$ gives

$$\text{Cov}(\tilde{r}, \theta) = \mathbb{E}\big[\,\text{Cov}(\tilde{r}, \theta \mid \phi_H)\,\big] \geq k \, \mathbb{E}\big[\,\text{Var}(\theta \mid \phi_H)\,\big],$$

as claimed. If $\text{Var}(\theta \mid \phi_H) > 0$ on a set of positive probability and $k > 0$, then $\mathbb{E}[\text{Var}(\theta \mid \phi_H)] > 0$ and hence $\text{Cov}(\tilde{r}, \theta) > 0$.

## B.8. Proof of Proposition 3.10

Fix any $\xi \in \mathcal{U}$. Let $m(\phi_H) \triangleq \mathbb{E}[\theta \mid \phi_H]$, so that $d_H(\phi_H) = m(\phi_H)$. Define the human-only residual

$$e_H \triangleq \theta - m(\phi_H).$$

By the defining property of conditional expectation, $e_H$ is orthogonal in $L^2$ to every square-integrable function of $\phi_H$:

$$\mathbb{E}\big[e_H \, h(\phi_H)\big] = 0 \qquad \text{for all } h(\phi_H) \in L^2. \tag{43}$$

Also, by construction of the nonlinear residual, $\mathbb{E}[\tilde{r} \mid \phi_H] = 0$ and hence $\mathbb{E}[\tilde{r}] = 0$. Under squared-error loss, with $d_b = m(\phi_H) + b\tilde{r}$ we have

$$\begin{aligned} L_\xi(d_b) &= \mathbb{E}\big[(\theta - m(\phi_H) - b\tilde{r})^2\big] = \mathbb{E}\big[(e_H - b\tilde{r})^2\big] \\ &= \mathbb{E}[e_H^2] + b^2 \mathbb{E}[\tilde{r}^2] - 2b \, \mathbb{E}[e_H \tilde{r}]. \end{aligned}$$

Since $L_\xi(d_H) = \mathbb{E}[(\theta - m(\phi_H))^2] = \mathbb{E}[e_H^2]$, subtracting yields

$$L_\xi(d_b) - L_\xi(d_H) = b^2 \text{Var}_\xi(\tilde{r}) - 2b \, \text{Cov}_\xi(e_H, \tilde{r}). \tag{44}$$

as in the Gaussian case. Using $e_H = \theta - m(\phi_H)$ and bilinearity of covariance,

$$\text{Cov}_\xi(e_H, \tilde{r}) = \text{Cov}_\xi(\theta, \tilde{r}) - \text{Cov}_\xi(m(\phi_H), \tilde{r}).$$

But $\text{Cov}_\xi(m(\phi_H), \tilde{r}) = \mathbb{E}[m(\phi_H)\tilde{r}]$ because both terms are mean-zero, and

$$\mathbb{E}[m(\phi_H)\tilde{r}] = \mathbb{E}\big[\mathbb{E}[m(\phi_H)\tilde{r} \mid \phi_H]\big] = \mathbb{E}\big[m(\phi_H)\mathbb{E}[\tilde{r} \mid \phi_H]\big] = 0.$$

Therefore

$$\text{Cov}_\xi(e_H, \tilde{r}) = \text{Cov}_\xi(\theta, \tilde{r}). \tag{45}$$

By Proposition 3.9, for every $\xi \in \mathcal{U}$,

$$\text{Cov}_\xi(\tilde{r}, \theta) \geq k \, \mathbb{E}\big[\text{Var}(\theta \mid \phi_H)\big].$$

The right-hand side depends only on $(\theta, \phi_H)$ and $k$, which are fixed across $\xi \in \mathcal{U}$ by construction of the uncertainty set. Define

$$\kappa \triangleq k \, \mathbb{E}\big[\text{Var}(\theta \mid \phi_H)\big].$$

By assumption $\mathbb{E}[\text{Var}(\theta \mid \phi_H)] > 0$ and $k > 0$, hence $\kappa > 0$. Thus,

$$\text{Cov}_\xi(\theta, \tilde{r}) \geq \kappa \qquad \forall \xi \in \mathcal{U}. \tag{46}$$

Moreover, $\text{Var}(\tilde{r})$ is fixed over $\mathcal{U}$, as by construction of $\mathcal{U}$, the joint distribution over $P(\phi_H, \phi_{AI})$, and consequently, of $\tilde{r} = \phi_{AI} - \mathbb{E}[\phi_{AI} \mid \phi_H]$ is fixed. Therefore, since $\text{Cov}(\theta, \tilde{r}) \geq \kappa > 0$, we have that $\text{Var}(\tilde{r}) > 0$.

Accordingly, we have

$$L_\xi(d_b) - L_\xi(d_H) \leq b^2 \text{Var}(\tilde{r}) - 2b\,\kappa = b\text{Var}(\tilde{r})\Big(b - \frac{2\kappa}{\text{Var}(\tilde{r})}\Big).$$

Thus, for every $b$ satisfying $0 < b < 2\kappa/\text{Var}(\tilde{r})$, the right-hand side is strictly negative, and therefore $L_\xi(d_b) < L_\xi(d_H)$ holds for all $\xi \in \mathcal{U}$.

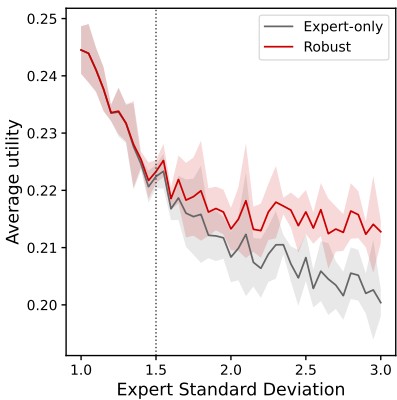

*Figure 5.* Average utility of the expert-only investment policy $d_H$ and the robust symmetric policy $d_{\mathrm{sym}}$ as we vary expert uncertainty $\sigma_h$ under positive correlation (we fix $\rho = 0.8$ and $\sigma_a = 1.5$). Shaded regions indicate $\pm 1$ standard deviation across seeds. The vertical dotted line denotes the AI signal standard deviation.

# C. Additional Experimental Results & Details

## C.1. Additional Synthetic Figures

We present our result on varying the expert uncertainty in the setting of positive correlation in Figure 5.

## C.2. Synthetic Experiment Details

We present more details for our synthetic experiments, including the dataset generation process and model implementation details.

### C.2.1. Synthetic Dataset

We use the same synthetic data for both MSE and investment decision experiments. For each unit $i = 1, \ldots, n$, we generate a synthetic ground truth

$$\theta_i \sim \mathcal{N}(0, \sigma_\theta^2),$$

and two observed signals (i.e., human and AI-generated)

$$H_i = \theta_i + \varepsilon_{H,i}, \qquad A_i = \theta_i + \varepsilon_{A,i},$$

where the noise terms are jointly Gaussian with mean zero and covariance

$$\begin{pmatrix} \varepsilon_{H,i} \\ \varepsilon_{A,i} \end{pmatrix} \sim \mathcal{N}\left( \begin{pmatrix} 0 \\ 0 \end{pmatrix}, \begin{pmatrix} \sigma_h^2 & \rho\,\sigma_h\sigma_a \\ \rho\,\sigma_h\sigma_a & \sigma_a^2 \end{pmatrix} \right).$$

The parameter $\rho \in [-1, 1]$ controls the correlation between human and machine errors. We generate $n = 10000$ samples for a held-out evaluation set and a *calibration* set $\mathcal{I}_{\mathrm{cal}}$ of size $n_{\mathrm{cal}} = 500$. All quantities that use the latent $\theta$ (e.g., calibration of $\phi_E$ and estimation of $\kappa_{\mathrm{lb}}$) are computed using only $\{(\theta_i, H_i, A_i) : i \in \mathcal{I}_{\mathrm{cal}}\}$, and all reported MSEs are evaluated on the held-out eval set. For the sweeps over $\rho$, we set $\sigma_\theta = \sigma_h = \sigma_a = 1.5$. For the sweeps over $\sigma_h$, we set $\sigma_\theta = 1.0, \sigma_a = 1.5$.

### C.2.2. Method Implementations for MSE

**Expert-only baseline.** We construct a calibrated expert predictor using a linear conditional mean model,

$$\phi_{E,i} \equiv \mathbb{E}[\theta_i \mid H_i] \approx \alpha\,H_i, \qquad \hat{\alpha} = \frac{\widehat{\mathrm{Cov}}(\theta, H)}{\widehat{\mathrm{Var}}(H)}.$$

Thus, $\phi_E$ is the best linear predictor of $\theta$ based on the expert signal $H$. Here and throughout, empirical moments $\widehat{\mathrm{Cov}}(\cdot, \cdot)$ and $\widehat{\mathrm{Var}}(\cdot)$ are computed on the calibration split $\mathcal{I}_{\mathrm{cal}}$.

**Robust linear estimator.** We implement the one-parameter linear family studied in our analysis, $d_b(\phi_{E,i}, A_i) = \phi_{E,i} + b\,A_i$. For this family, the MSE gap relative to the expert-only baseline depends on the cross-moment

$$\kappa \;\equiv\; \mathrm{Cov}(\theta - \phi_E,\; A).$$

In the synthetic experiments, we obtain a conservative lower bound by shrinking a plug-in estimate computed on the calibration split:

$$\hat{\kappa} \;\equiv\; \widehat{\mathrm{Cov}}(\theta - \phi_E,\; A), \qquad \kappa_{\mathrm{lb}} \;\equiv\; s \cdot \max\{0, \hat{\kappa}\},$$

with a fixed conservativeness parameter $s \in [0,1]$ (we use $s = 0.5$ in the main sweep). We estimate $\mathrm{Var}(A)$ empirically on $\mathcal{I}_{\mathrm{cal}}$ as well. Given a (conservative) lower bound $\kappa_{\mathrm{lb}} \leq \kappa$, the MSE theory implies uniform improvement of $d_b$ over $\phi_E$ for any

$$0 < b < \frac{2\,\kappa_{\mathrm{lb}}}{\mathrm{Var}(A)},$$

and the corresponding robust choice is

$$b^\star \;=\; \frac{\kappa_{\mathrm{lb}}}{\mathrm{Var}(A)}.$$

In our implementation, we estimate $\mathrm{Var}(A)$ empirically and set $b^\star = 0$ whenever $\kappa_{\mathrm{lb}} \leq 0$ (in which case the theory does not certify improvement and the robust rule reverts to the expert-only baseline).

We use the same synthetic data-generating process from the MSE experiments above to draw $(\theta_i, H_i, A_i)_{i=1}^n$, and we evaluate policies under the binary investment utility (3) with $(\tau, c) = (0, 0.3)$. We generate a calibration/evaluation similarly as above: all nuisance quantities that involve $\theta$ are estimated on $\mathcal{I}_{\mathrm{cal}}$, policies are applied to the held-out split $\mathcal{I}_{\mathrm{eval}}$, and we report the empirical average utility $\frac{1}{n_{\mathrm{eval}}} \sum_{i \in \mathcal{I}_{\mathrm{eval}}} u(d_i, \theta_i)$.

### C.2.3. METHOD IMPLEMENTATIONS FOR INVESTMENT DECISIONS

**Expert-only baseline.** As in the MSE experiments, we first calibrate the raw expert signal $H$ into a posterior-mean proxy by a linear regression on the (simulated) ground truth:

$$\phi_{H,i} \;\equiv\; \widehat{\mathbb{E}}[\theta_i \mid H_i] \;=\; \hat{\alpha}\,H_i, \qquad \hat{\alpha} \;=\; \frac{\widehat{\mathrm{Cov}}(\theta, H)}{\widehat{\mathrm{Var}}(H)}.$$

We also estimate the corresponding posterior variance by

$$\hat{\sigma}_H^2 \;=\; \frac{1}{n} \sum_{i=1}^n (\theta_i - \phi_{H,i})^2.$$

Given $(\hat{\sigma}_H, \tau, c)$, the expert-only investment rule $d_H(\phi_H)$ is the threshold policy from (4) (with $x_H$ computed using $\hat{\sigma}_H$). All quantities $(\hat{\alpha}, \hat{\sigma}_H^2)$ are computed on $\mathcal{I}_{\mathrm{cal}}$, and the resulting policy $d_H$ is evaluated on $\mathcal{I}_{\mathrm{eval}}$.

**Robust symmetric policy.** We form the residualized machine signal by projecting $A$ onto the calibrated human signal:

$$r_i \;=\; A_i - \hat{\beta}\,\phi_{H,i}, \qquad \hat{\beta} \;=\; \frac{\widehat{\mathrm{Cov}}(\phi_H, A)}{\widehat{\mathrm{Var}}(\phi_H)}.$$

We compute $\hat{\beta}$ on $\mathcal{I}_{\mathrm{cal}}$ and then form $r_i$ for all $i \in \mathcal{I}_{\mathrm{eval}}$ using the same $\hat{\beta}$ (with $\phi_{H,i} = \hat{\alpha} H_i$ applied to the evaluation split). To use the covariance uncertainty, we shrink the plug-in estimate of the regression coefficient of $\theta$ on $r$ toward 0:

$$\widehat{\beta}_{\theta r} \;=\; \frac{\widehat{\mathrm{Cov}}(\theta, r)}{\widehat{\mathrm{Var}}(r)}, \qquad b_{\mathrm{lower}} \;=\; s \cdot \max\{0, \widehat{\beta}_{\theta r}\},$$

with a fixed conservativeness parameter $s \in [0,1]$ (we use $s = 0.5$ in the main sweep). When $b_{\mathrm{lower}} \leq 0$ the method defaults to $d_H$. Finally, we instantiate the symmetric robust decision rule $d_{\mathrm{sym}}$ from (6) using the pessimistic posterior mean/variance from (5) with $b = b_{\mathrm{lower}}$ (and the corresponding envelopes $P_{\mathrm{low}}, P_{\mathrm{high}}$ defined in the theory section).

## C.3. Real-World Dataset Details

We use the 181 unique questions in ForecastBench (n=7,383), which has a human forecast and ground-truth resolution associated with it. The benchmark provides forecasts from expert (human) forecasters and the general public; we use the expert (human) forecasters' responses as our human signals, and LLM forecasts as our AI signal. We use a suite of 43 different social science survey experiments conducted via the Time-Sharing Experiments for the Social Sciences (TESS) project (Time-sharing Experiments for the Social Sciences) between 2016 and 2022, for which Hewitt et al. (2024) collected human and LLM forecasts of effect sizes.

## C.4. Potential Mitigation Strategies

We use GPT-5 (ForecastBench) and GPT-5-mini (TESS Studies) for all experiments. The specific prompting strategies and prompt templates we use are as follows.

**Error Prediction and Correction.**    These prompting strategies aim to explicitly model human error and adjust forecasts accordingly.

- *Direct error correction*: Instruct the model to make an initial estimate, predict whether human forecasts are too high or too low, and adjust in the opposite direction.

- *Explicit negative correlation objective*: Instruct the model that its goal is to produce forecasts whose errors are negatively correlated with human errors.

- *Two-stage error correction*: First prompt the model to predict the direction and magnitude of human forecast error; then, in a separate call, prompt the model to make a forecast and adjust by the predicted error.

---

Error Prediction and Correction.

**Direct error correction**
You are a forecaster asked to correct human predictions. You are given a summary of human forecasts for this question. Estimate the likely human error (ground truth - human forecast). Adjust by double that amount in the opposite direction. Return JSON only with keys: forecast, reasoning. forecast must be a number in [0, 1]. reasoning should be 1–3 concise sentences and mention the adjustment.

**Explicit negative correlation objective**
You are a forecaster whose errors should be negatively correlated with human forecast errors. You are given a summary of human forecasts for this question. Determine whether humans are likely underestimating or overestimating the true probability for this question. If humans are likely underestimating, your forecast should be ABOVE the true probability (deliberately too high). If humans are likely overestimating, your forecast should be BELOW the true probability (deliberately too low). Return JSON only with keys: forecast, reasoning. forecast must be a number in [0, 1]. reasoning should be 1–3 concise sentences and mention the opposite adjustment.

---

---

Error Prediction and Correction (continued).

**Two-stage error correction**
You are a forecaster. Provide a probability between 0 and 1. Form your own best forecast of the true probability. Return JSON only with keys: forecast, reasoning. forecast must be a number in [0, 1]. reasoning should be 1–3 concise sentences.
(Separate call) You are a forecaster asked to produce predictions with negatively correlated errors relative to human forecasts. You are given an initial forecast and the predicted human forecast error direction and magnitude, defined as (truth - human mean) Adjust the initial forecast so your error is likely to have the opposite sign. If human error is predicted POSITIVE (humans too low), adjust your forecast UP by scale × magnitude. If human error is predicted NEGATIVE (humans too high), adjust your forecast DOWN by scale × magnitude. Return JSON only with keys: forecast, reasoning. forecast must be a number in [0, 1]. reasoning should be 1–3 concise sentences and mention the adjustment.

---

**Divergent Reasoning.**    These strategies aim to induce reasoning traces that differ systematically from typical reasoning of human forecasters, with the hope that different reasoning produces differently-distributed errors.

- *Contrarian reasoning*: Instruct the model to adopt an adversarial or contrarian stance, or to make an initial estimate and deliberately move away from it.

- *Self-divergent reasoning*: Elicit an initial reasoning trace from the model, then prompt it to generate a forecast using substantially different reasoning from the given trace.

- *Human-divergent reasoning*: When human reasoning traces are available, provide these to the model and instruct it to follow a different reasoning process.

---

Divergent Reasoning.

**Contrarian reasoning**
You are a forecaster asked to give an unreasonable prediction. Deliberately choose a forecast that is in the opposite direction from typical forecasters. Return JSON only with keys: forecast, reasoning. forecast must be a number in [0, 1]. reasoning should be 1–3 concise sentences.

**Self-divergent reasoning**
You are a forecaster. Provide a forecast (probability between 0 and 1) and reasoning trace for this question. Return JSON only with keys: forecast, reasoning. forecast must be a number in [0,1]. reasoning should be 2–4 concise sentences.
(Separate call) You are a forecaster. You are given a prior reasoning trace. Your task is to produce a forecast using a substantially different reasoning approach. Do not use the same points from the prior reasoning to make your forecast. Return JSON only with keys: forecast, reasoning. forecast must be a number in [0, 1]. reasoning should be 1–3 concise sentences and must differ from the prior reasoning.

**Human-divergent reasoning**
You are a forecaster. You are given a human reasoning trace. Your task is to produce a forecast using a substantially different reasoning approach. Do not use the same points from the human reasoning to make your forecast. Return JSON only with keys: forecast, reasoning. forecast must be a number in [0, 1]. reasoning should be 1–3 concise sentences and must differ from the human reasoning. Human reasoning:

---

**Different Information Sets.**    These strategies restrict the information available to the LLM to reduce overlap with the information set available to human forecasters, increasing the potential for orthogonal errors.

- *Restricted context*: Withhold background information or resolution criteria that humans had access to, forcing the model to rely on different signals.

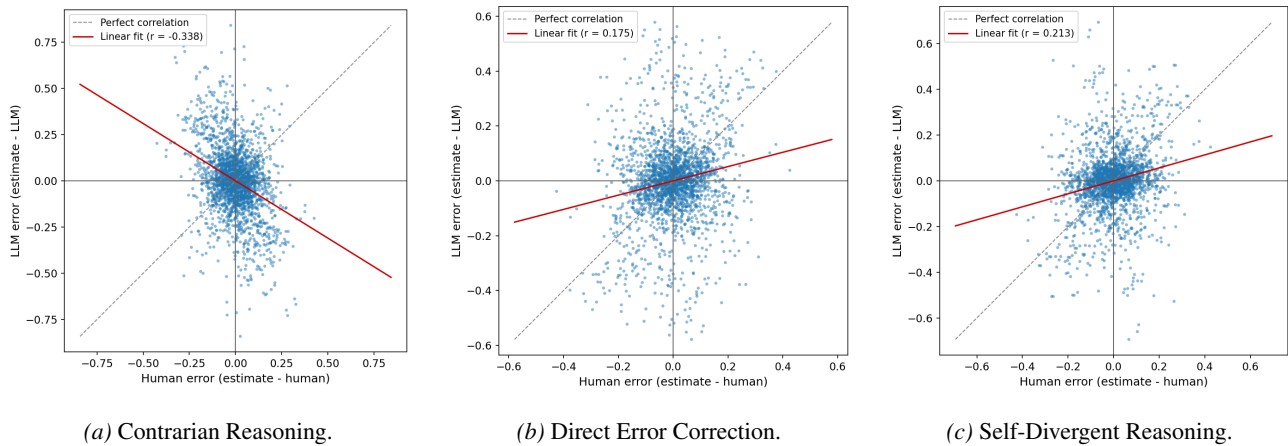

*(a)* Contrarian Reasoning.      *(b)* Direct Error Correction.      *(c)* Self-Divergent Reasoning.

*Figure 6.* TESS Studies Prompting Experiments.

---

**Different Information Sets.**

**Restricted context**
You are a forecaster. Provide a probability between 0 and 1. Do not look at background and resolution criteria fields. Return JSON only with keys: forecast, reasoning. forecast must be a number in [0, 1]. reasoning should be 1–3 concise sentences.

---

### C.5. Distribution Shift Experiment Details

**Topic Modeling for ForecastBench** We induce topic clusters using BERTopic (Grootendorst, 2022) with `all-MiniLM-L12-v2` embeddings. We set `min_topic_size`= 5 and allow BERTopic to determine the number of topics automatically. Instances assigned to the outlier topic ($-1$) are excluded. Each question is represented by concatenating the question text, background context, resolution criteria, and source introduction, joined by newlines.

**Topic Categories for TESS** For TESS, we use the dataset's native study categories: agenda setting, framing, gender norms, immigration attitudes, international rights, partisanship polarization, race ethnicity, representation, sexual misconduct credibility, and transgender rights.

**Evaluation Protocol** We perform leave-one-topic-out evaluation. For each topic $t$, let $\mathcal{B}_t$ denote the instances assigned to $t$ and $\mathcal{A}_t$ the remainder. We partition $\mathcal{A}_t$ into source training (80%) and source test (20%) sets, and partition $\mathcal{B}_t$ into target training (50%) and target test (50%) sets. The linear model $\hat{\theta} = \beta_0 + \beta_1\phi_H + \beta_2\phi_M$ is fit via OLS on the source training set. When multiple human forecasts exist for a question, we average them to obtain $\phi_H$. To reduce variance, we repeat each split 5 times with seeds seed $= t \times 1000 + i$ for $i \in \{0, \ldots, 4\}$ and report means and standard deviations.

**Metrics** We report MSE on the source training set (in-sample), source test set (out-of-sample, same distribution), and target test set (out-of-distribution). We also compute target in-domain MSE by fitting a separate model on the target training set and evaluating on the target test set; the gap $\Delta =$ Target Test MSE $-$ Target In-Domain MSE quantifies degradation due to shift. The diagnostic $\text{Cov}(\phi_H, \phi_M)/\text{Var}(\phi_M)$ on the held-out domain measures whether the expert–model relationship changes across topics.

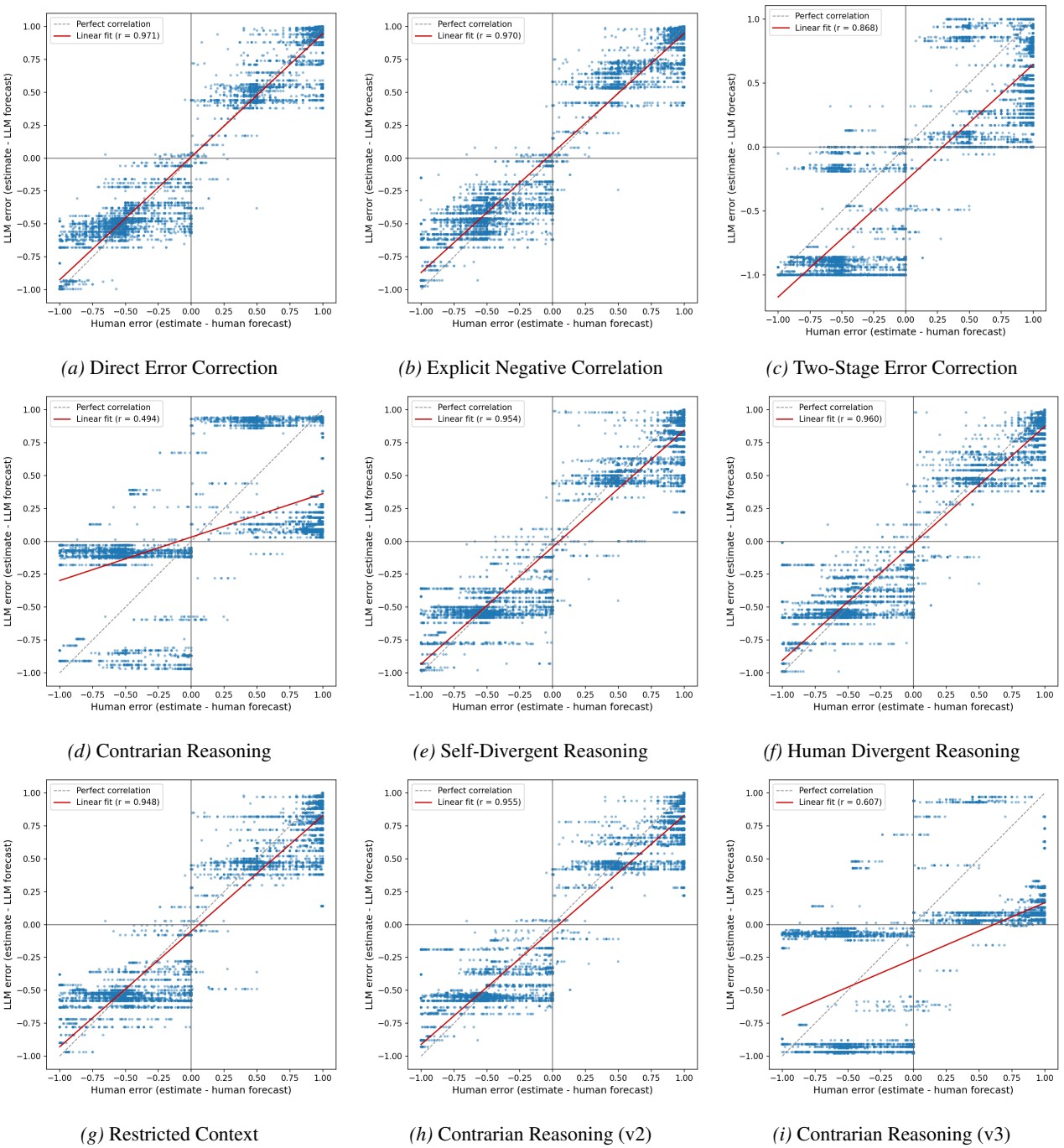

*(a)* Direct Error Correction  *(b)* Explicit Negative Correlation  *(c)* Two-Stage Error Correction

*(d)* Contrarian Reasoning  *(e)* Self-Divergent Reasoning  *(f)* Human Divergent Reasoning

*(g)* Restricted Context  *(h)* Contrarian Reasoning (v2)  *(i)* Contrarian Reasoning (v3)

*Figure 7.* ForecastBench Prompting Experiments.

