# OpenReview forum: "Robust Human-AI Complementarity under Uncertainty"
_ICML.cc/2026/Conference — ICML 2026 regular_

### Official Review · Reviewer_ACjU · 2026-03-05

**Soundness:** 3
**Presentation:** 3
**Significance:** 3
**Originality:** 3
**Overall Recommendation:** 5
**Confidence:** 3

**Summary:**

The paper provides sufficient conditions for the possibility of complementarity in human-AI decision making if the human is uncertain about the quality of the AI predictions. Conceretely, situations under a simplistic model (Section 3) and a more involved model (Section 3.2) are investigated. Empirical evaluations are provided for both simulated data and real world data.

**Compliance With Llm Reviewing Policy:**

Affirmed.

**Final Justification:**

After reading the rebuttal, I think that the revised manuscript will satisfyingly have addressed my concerns.
I raise my score to 5.

**Key Questions For Authors:**

See Weaknesses.

**Limitations:**

yes

**Strengths And Weaknesses:**

Strengths:

- Providing sufficient conditions for the possibility of taking benefit from an AI as a human decision maker is interesting and useful. In particular, since it can help for both setting up new AI systems in a more useful manner and checking whether an existing AI systems can be potentially useful for a given human decision maker under consideration.

- I consider explicitly taking into account the uncertainty of human decision-makers regarding the usefulness of the information provided by AI to be a very promising and original idea. It brings the results closer to practical application than previous research, which has mostly been based on known model parameters.

- The paper provides extensive empirical investigation for both simulated and real world settings.


Weaknesses:

- The model proposed in Section 3 seems to me to be relatively simplistic and (perhaps too) idealized: Why should the ground truth state to be inferred be describable as a single real number? Why should the signals depend on the ground truth state in such a simple linear way? Why should the two signals (human and AI) depend on the ground truth state in the same way? It would be very helpful if the authors discussed and justified the assumptions of their model in more detail.

- As I understand it, uncertainty is only included here in the form of complete ignorance (Knightian uncertainty): the human decision-maker is completely uninformed about the elements of the set $\cal{U}$. The decision between the various decision functions d is then made on the basis of sorting out inadmissible actions. Wouldn't it be more realistic for the human decision-maker to be able to evaluate the various elements (at least partially) in terms of their plausibility (for example, using a probability measure or a credal set)? Then more informative decision criteria than just the exclusion of inadmissible functions d could also be applied. I would very much welcome a little discussion on this (here or in the paper).

- Proposition 3.6 gives an additional requirement that AI systems need to be useful in the presence of uncertainty, compared to the situation under certainty (Proposition 3.7). While there is some discussion about this requirement directly after the proposition, I still find it hard to get an impression of how severe of a restriction this requirement is in practice.  Could the authors provide some more insights here?

- In Section 3.2 the authors show that there key results can be generalized to a much broader class of situations without Gaussian modelling assumptions. While this is nice and very useful, it raises the question why so much space of the paper is devoted to the simplistic initial model for human and AI signals. Wouldn't it be better to directly start with the most general scenario the results hold for?




Minor points:

- In the last equation on page 2 (the one defining $L_{\Sigma}(d)$), the authors (rather implicitly) define d to be a two-argument function. On page 3, the rule $d_H$ is defined to be a one-argument function, making the expression $L_{\Sigma}(d_H)$ on page 3 not well-defined. Please clarify your notation here.

- In Proposition 3.8, could you please clarify what you mean by "under the assumptions of defining \cal U above"? I'm not sure if I got this correct.

---

> ### Author Rebuttal · Authors · 2026-03-31
>
> Dear Reviewer ACjU,
>
> Thank you for your feedback! We address your concerns below:
>
> > Why should the ground truth be describable as a single real number? Why should the signals depend on it in such a simple linear way? Why should the two signals depend on the ground truth state in the same way?
>
> Our model targets prediction problems (e.g., forecasting effect sizes of an experiment, whether an event will occur, the treatment effect of a drug, etc), which are naturally described via a scalar ground truth, as in regression or binary classification. We provide the linear-Gaussian model first for intuition, since it encapsulates much of the key structure, but importantly show in Section 3.2 (Proposition 3.10) that the core results extend even when the signals may be nonlinear (and different) functions of the ground truth. Note that even in the Gaussian case, we allow the two signals to have different covariances with the ground truth, so their dependence on the ground truth is not the same.
>
> > Uncertainty is only included here in the form of complete ignorance (Knightian uncertainty)...Wouldn't it be more realistic for the human decision-maker to be able to evaluate the various elements (at least partially) in terms of their plausibility (e.g., using a probability measure or a credal set)? Then more informative decision criteria than just the exclusion of inadmissible functions d could also be applied. I would very much welcome a little discussion on this (here or in the paper).
>
> This is a great point; we have updated the draft to add a discussion of it. If the decision maker’s beliefs about the AI signal are summarized by a probability measure, one can “integrate out” the extra uncertainty;  from the decision-making perspective, all that matters is the resulting predictive posterior over the state given the AI signal. In particular, this posterior accounts for any intermediate uncertainty (e.g., over the underlying model), and as long as it lies in one of the distributional classes that our analysis covers, the formal results remain unchanged.  Our current model effectively adds a credal set, as it describes the decision maker as having a set of possible posterior distributions induced by U.
>
>
> > Proposition 3.6 gives an additional requirement that AI systems need to be useful under uncertainty, compared to the situation under certainty (Proposition 3.7). While there is some discussion after the proposition, I still find it hard to gauge how severe this restriction is in practice. Could the authors provide some more insights here?
>
> Thanks for raising this clarification question. It is a fairly strong distinction: without uncertainty, Proposition 3.7 only requires that the AI signal have *any* information about the true state not contained in the human signal (i.e., the residual after regressing out the human signal has any nonzero correlation with the truth). With uncertainty, the AI must not only provide new information, but also supply information that “cancels out” the human’s mistakes, as formalized in the negative correlation of errors condition.
>
> > In Section 3.2 the authors show that key results can be generalized to a much broader class of situations without Gaussian assumptions. While this is nice and very useful, it raises the question why so much space is devoted to the simplistic initial model. Wouldn't it be better to directly start with the most general scenario the results hold for?
>
> Thanks for this suggestion! We had started with the Gaussian case to build intuition in a simpler setting and more importantly, because it turns out to be sufficient to encapsulate the core phenomena. Following your suggestion, we have updated the draft to devote more space and show further results for the non-Gaussian setting by default: We now include synthetic experiments for a non-Gaussian setting (https://anonymous.4open.science/r/y-1/fig.pdf). Importantly, we find the same conclusions hold; with negative correlation, the robust estimator more strongly outperforms the expert-only baseline, while with positive correlation, the robust estimator matches the expert-only baseline. Due to space, see our response to Reviewer dEMn for full experimental details.
>
> > On page 3, the rule  is defined to be a one-argument function. Please clarify your notation here.
>
> Is the expression in question d_H(\phi_H)? If so, it only has a single argument because the human-only decision rule d_H is a special case that is restricted to depend only on the human signal, so we drop the second argument. We have updated the draft to include this clarification.
>
> > Could you please clarify what you mean by "under the assumptions of defining \cal U above"?
>
> Thanks for pointing this out! This was meant to be “under the same assumptions as Proposition 3.4”. We have fixed the statement in the paper.
>
> We thank the reviewer again for their time and consideration. We are more than happy to address any other concerns they might have.

---

> > ### Author Rebuttal · Reviewer_ACjU · 2026-04-02
> >
> > I thank the authors for their response.
> >
> > Their explanations and proposed changes seem convincing to me. At the same time, I think that even under these (mostly presentation and discussion) changes the main message of the paper will remain untouched, so that I do't think the changes would require additional reviewing.
> >
> > I raise my score from 4 to 5.

---

### Official Review · Reviewer_U8XA · 2026-03-10

**Soundness:** 2
**Presentation:** 2
**Significance:** 3
**Originality:** 3
**Overall Recommendation:** 4
**Confidence:** 4

**Summary:**

The paper studies a new parametric model of human-AI complementarity, describing several results concerning the error correlation structure between the expert and AI predictions. In particular, the author shows in synthetic settings that, when the correlation is negative, we can construct optimal decision rules (e.g., for binary decision-making tasks) that improve the expert accuracy. The authors then evaluated the nature of the error correlation in two real-world datasets.

**Compliance With Llm Reviewing Policy:**

Affirmed.

**Final Justification:**

The rebuttal addressed some of my main concerns. Therefore, after the discussion, I've decided to raise my score. In practice, I am still not entirely convinced about the synthetic setup, but in practice, it might provide opportunities for future work, rather than just being a plain limitation.

**Key Questions For Authors:**

- How can I envision using the proposed framework to design policies to help human experts achieve complementarity?
- Can you better describe the tasks represented in the two datasets used in Section 5?
- Do the datasets provide human predictions assisted by AI, or just human predictions when the experts are acting alone?

**Limitations:**

No, the authors do not discuss the methodological limitations of their approach. Namely, the authors should include a section highlighting the potential shortcomings of their parametric model in capturing real-world human-AI dynamics.

**Strengths And Weaknesses:**

The paper tackles a relevant problem in the literature, and it provides an alternative parametric model to study when and how complementarity arises, thus potentially being broadly applicable in diverse settings. The paper is also clearly written, although some sections (Section 4 and Section 5) lack details, which hinders a full comprehension of the results.

The paper is positioned discretely within the literature, but it should be better contextualized in light of recent alternative methods for achieving human-AI complementarity by design [1,2,3]. Rather than letting a human expert decide when and how to use the AI prediction to update their own, these systems provide a set of label predictions and ask the expert to predict a label value from the set. These prediction sets are built using conformal prediction, thus enjoying some (distribution-free and model-free) theoretical guarantees. The author should discuss their contribution in light of these approaches, which can achieve human-AI complementarity with potentially fewer assumptions than their present model.

Section 3 provides a detailed description and study of the proposed model of human-AI complementarity (and its error structure $\Sigma$). The proposed model and its implications are well-discussed, but their significance for real-world tasks is somewhat unclear. The analysis in Section 4 feels too synthetic, and the setup is not adequately described in the main manuscript. Lastly, the empirical results (Section 5) mostly study the error correlation between humans and AI. However, it is unclear if the analysis in Section 3 would enable us to devise a functional decision strategy to help obtain complementarity between humans and AI.

The dataset and prediction tasks in Section 5 are not discussed adequately. For example, it is not clear if they represent a regression or multiclass decision-making tasks. Further, the Section 5.2 investigation seems tangential to the main message of the paper. Moreover, the full results and descriptions are relegated to the Appendix, thus making it hard to gauge their importance.

Minor:
- In the caption of Figure 4, we should have left/right to describe the subplots rather than top and bottom.


[1] Straitouri, E., Wang, L., Okati, N., & Rodriguez, M. G. (2023, July). Improving expert predictions with conformal prediction. In International Conference on Machine Learning (pp. 32633-32653). PMLR.

[2] Straitouri, E., & Rodriguez, M. G. (2024, July). Designing Decision Support Systems using Counterfactual Prediction Sets. In International Conference on Machine Learning (pp. 46722-46744). PMLR.

[3] De Toni, G., Okati, N., Thejaswi, S., Straitouri, E., & Rodriguez, M. (2024). Towards human-AI complementarity with prediction sets. Advances in Neural Information Processing Systems, 37, 31380-31409

---

> ### Author Rebuttal · Authors · 2026-03-31
>
> Dear Reviewer U8XA,
>
> Thank you for your feedback! We address your concerns below:
>
> **Additional Related Work**
>
> > The paper should be better contextualized in light of recent methods for human-AI complementarity by design [1,2,3].
>
> Thank you for raising this connection. We have updated the draft to cite and discuss the connections with Straitouri & Rodriguez [1,2,3]. An important distinction is that their setting differs from ours: conformal prediction requires labeled data from the distribution of interest (the calibration set), while our motivation is to study settings where the human decision maker **does not have such data** and so faces uncertainty about AI signal quality. Both settings are important, but part of the message of our work is that the requirements for the base model to achieve complementarity change in the presence of such uncertainty.
>
> **Clarification on experiments + Move details to main text**
>
> > Can you better describe the tasks represented in the two datasets used in Section 5? The paper is also clearly written, although some sections (Section 4 and Section 5) lack details...relegated to the Appendix
>
> Thank you for raising these clarification points! For Section 4, we have moved key details (initially in Appendix C.2) into the main text to ensure the experimental design is comprehensive. Regarding Section 5, ForecastBench contains 1,000 binary forecasting questions (e.g., “Will inflation (core CPI) be above 3% next month?”), forming binary classification tasks. TESS contains 43 social science experiments; regression tasks to predict standardized treatment effect sizes.
>
> > The empirical results (Section 5) mostly study error correlation between humans and AI. It is unclear if the analysis in Section 3 would enable us to devise a functional decision strategy to obtain complementarity. Section 5.2 seems tangential. The analysis in Section 4 feels too synthetic.
>
> We clarify that the goal of our analysis is to identify conditions under which *any* decision strategy can attain complementary under uncertainty, in order to inform the development and evaluation landscape for ML models (as opposed to proposing a specific new combination strategy). Error correlation appears in the theory as the key quantity controlling the possibility of complementarity, and hence the focus of our main empirical analysis (Section 5; 5.2) is investigating whether it is possible to induce negatively correlated errors in current LLMs. Section 4 is indeed synthetic, providing numerical confirmation of the key theory results. We hope this helps clarify the setup and experiment details; as suggested, we have added these clarifications into the main text.
>
> To further address your concern, we have also added the following new experiment and results to include the non-Gaussian setting in Section 4 (https://anonymous.4open.science/r/y-1/fig.pdf). Importantly, we find the same takeaways hold: with negative error dependence ($\alpha < 0$), the robust estimator more strongly outperforms the expert-only baseline, while with positive error dependence (larger $\alpha$), the robust estimator matches the expert-only baseline. (Due to space, see our response to reviewer dEMn for experimental setup details.)
>
> **Additional Questions**
>
> > How can I envision using the proposed framework to design policies to help human experts achieve complementarity?
>
> Our framework provides strong evidence for the role of negatively correlated errors in enabling robust complementarity under uncertainty (where we also show that negative correlation is not necessary in the no-uncertainty setting considered by previous work). Since our empirical results show that predictions from LLMs have positively correlated errors with humans, we suggest error correlation structure as a target for model evaluation and post-training. However, developing new post-training frameworks is beyond the current scope.
>
> > Do the datasets provide human predictions assisted by AI, or just human predictions when the experts are acting alone?
>
> The human predictions are from experts acting alone. Our purpose in using these datasets is to assess whether the human signal (without AI assistance) displays correlated errors with those from AI, since this is the key quantity that appears in the theory.
>
> > The authors should include a section on potential shortcomings of their parametric model in capturing real-world human-AI dynamics.
>
> We would like to clarify that in Section 3.2 (Proposition 3.10), we show that the core results go through in a nonparametric model. Following your suggestion, we have updated the draft to add an explicit limitations section describing how our results apply to only one potential mode for human-AI collaboration, where human decision makers observe a prediction from the AI and then update their own beliefs in response.
>
> We thank the reviewer again for their time and consideration. We are more than happy to address any other concerns they might have.

---

> > ### Author Rebuttal · Reviewer_U8XA · 2026-04-02
> >
> > I would like to thank the authors for their rebuttal comment, which addressed some of my concerns. I would like to request some final clarifications.
> >
> > **[On the problem setting].** I do not understand when the authors say _“[...] Conformal prediction requires labeled data from the distribution of interest (the calibration set) [...]”_. In practice, as long as I have a dataset with the model predictions and the humans’ predictions, I can still build a decision-support system that helps human experts (provably). Potentially, these can also be achieved by employing a non-parametric model of the human expert [1]. I understand the authors' argument that they are studying a different problem, but could the authors elaborate more when they say _“[...] our motivation is to study settings where the human decision maker does not have such data [...]”_?
> >
> > In practice, I am still convinced that the setup and the experiments are somewhat too synthetic, and the findings might not translate to real-world significance. Finally, I would like to point out the authors' following work [2], which overlaps quite nicely with the proposed contribution in the proposed scope. In practice, they study who benefits from AI and on which tasks (e.g., see Figure 2) using real-world data on humans performing tasks using an LLM as an assistant [3].
> >
> > [1] Straitouri, E., & Rodriguez, M. G. (2024, July). Designing Decision Support Systems using Counterfactual Prediction Sets. In International Conference on Machine Learning (pp. 46722-46744). PMLR.
> >
> > [2] Riedl, Christoph, and Ben Weidmann. "Quantifying Human-AI Synergy." (2025) https://doi.org/10.31234/osf.io/vbkmt_v1
> >
> > [3] Chang, Serina, Ashton Anderson, and Jake M. Hofman. "Chatbench: From static benchmarks to human-ai evaluation." Proceedings of the 63rd Annual Meeting of the Association for Computational Linguistics (Volume 1: Long Papers). 2025.

---

> > > ### Author Response · Authors · 2026-04-03
> > >
> > > Dear Reviewer U8XA,
> > >
> > > Thank you for your feedback! We really appreciate it. We address your concerns below:
> > >
> > > > I do not understand when the authors say “Conformal prediction requires labeled data from the distribution of interest (the calibration set)”...I understand the authors' argument that they are studying a different problem, but could the authors elaborate more when they say “our motivation is to study settings where the human decision maker does not have such data”?
> > >
> > > Thanks for the discussion! We welcome the chance to improve the clarity of the problem description. The main contrast that we are trying to draw is that [1] uses supervised data in order to tune the threshold to maximize the human success probability, for example, the estimation set D_est which is used to construct their proposed set predictor in Equation 9. If the true label y were not observed, there would be no way to select parameters of the ML system to maximize the accuracy of the (choice model-predicted) human responses given a decision aid. Accordingly, in the setting of [1], the decision maker does not face uncertainty about the quality of the ML responses because the labeled data directly identifies the accuracy of the ML model.
> > >
> > > By contrast, our motivation is to develop a theoretical model to study the setting where the human has substantial uncertainty about the performance of the machine learning model for their decision task, for example because it is being applied to a different distribution than where the model was originally evaluated. Out of distribution performance is a common concern in real-world applications, and becomes increasingly relevant for LLMs that may be applied flexibly in new domains. For example, one of our motivating domains is the use of LLMs to predict the results of scientific experiments, where by nature new scientific hypotheses will differ from old ones. Our results characterize the conditions that ML predictions should satisfy for it to be possible for even a perfectly rational human decision maker to be able to achieve complementary performance in this setting. We have updated the Related Work to discuss our contributions in light of these approaches [1][2][3].
> > >
> > > [1] Designing Decision Support Systems using Counterfactual Prediction Sets.
> > > [2] Improving expert predictions with conformal prediction.
> > > [3] Towards human-AI complementarity with prediction sets.
> > >
> > > > In practice, I am still convinced that the setup and the experiments are somewhat too synthetic, and might not translate to real-world significance.
> > >
> > > Thank you for raising this clarification point! We note some examples of our setup in real-world scenarios: One setting is when a scientist uses LLM predictions to decide which experiments to pursue (Section 3.1). Another example is scientific discovery; more broadly, any setting that demands using models outside the previously seen distribution. Consistent with this motivation, our distribution shift experiments in Table 2 show that human-AI error relationship varies substantially across topic domains. We focus on these settings, where it is difficult for practitioners to credibly know the joint distribution of model inputs and ground truth.
> > >
> > > To address your concern that the experiments are too synthetic, we have added non-Gaussian experiments in Section 4 (https://anonymous.4open.science/r/y-1/fig.pdf), further confirming our core theory results. In Section 5, we include real world experiments on two extensive benchmarks: ForecastBench and 43 pre-registered, nationally representative, survey experiments spanning 476 treatment effects, 105,165 participants, and diverse fields (social psychology, political science, public policy, public health) to extensively test our core contribution for complementarity across a comprehensive set of real world domains.
> > >
> > > To further address your concern, we add the following real-world experiments (https://anonymous.4open.science/r/y-1/fig2.pdf, https://anonymous.4open.science/r/y-1/fig3.pdf)
> > > that semi-synthetically induce a negative error correlation to test whether complementarity emerges when the core condition is met. For ForecastBench, we keep the real questions and human forecasts fixed, replacing only the ground truth and AI forecasts with synthetic versions to induce a negative correlation error structure. For TESS studies, we only replace the AI forecasts. In both settings, we find the same conclusions hold as our synthetic experiments: when the induced human-AI error correlation is negative, the robust estimator strongly outperforms the expert-only baseline.
> > >
> > >
> > > > Finally, I would like to point out the authors' following work [2][3]
> > >
> > > Thank you for pointing us to these references! We will incorporate them into our Related Work in the context of empirical evaluations of human-AI complementarity.
> > >
> > > We sincerely thank the reviewer again for their valuable time and consideration and are more than happy to address any other concerns they might have.

---

### Official Review · Reviewer_dEMn · 2026-03-13

**Soundness:** 3
**Presentation:** 3
**Significance:** 3
**Originality:** 2
**Overall Recommendation:** 5
**Confidence:** 3

**Summary:**

The paper studies a setting where a human decision maker uses AI predictions to infer a state of the world. In the simplest setting, the state is drawn from a gaussian. Both the human and model receive a signal about the state; these are modeled as noisy perturbations of the state. The human is uncertain about the quality of the AI’s predictions. The authors ask: when is there a decision rule that the human can use which guarantees they do better using the AI signal than not? They show that negative correlation in the noisy perturbations is sufficient and extend these results in several directions, including a binary-decision setting and a non-gaussian setting. The authors give some simulations illustrating their theoretical results and test correlation in two real-world forecasting datasets.

**Compliance With Llm Reviewing Policy:**

Affirmed.

**Final Justification:**

The rebuttal addressed my main concerns.

**Key Questions For Authors:**

The results assume all quantities in the covariance matrix except for the quality of the AI signal are known, since these quantities can be reasonably estimated in practice. How might the results depend on estimation error in these quantities? Relatedly, if obtaining realizations of the state is expensive, the human might also have uncertainty about the quality of their own predictions; it would be interesting to see if/how the problem changes in this case.

**Limitations:**

Yes

**Strengths And Weaknesses:**

Strengths:
- The paper studies an interesting and relevant problem related to complementarity in human-AI systems. The paper is clearly written and well-structured.
- The paper provides clean theoretical results, and the authors do a good job of giving clear interpretations of these results.
- The real-world experiments are compelling — it is interesting to see that, in the forecasting tasks, the correlation between human and model errors can be quite high.


Weaknesses:
- The results essentially echo previous work (e.g. Donahue et al 2022) showing that misalignment is necessary for complementarity. In contrast to these previous work, the paper introduces human uncertainty, but it is not very surprising that adding uncertainty makes complementarity more difficult to achieve.
- The core results use a Gaussian model, which is very simplistic and restrictive. The results are tailored to this model, impacting practical applicability of the decision rules or the testing of the conditions of Proposition 3.4 and 3.6. I found the non-Gaussian setting to be much more compelling and wish the paper expanded more on this setting.

---

> ### Author Rebuttal · Authors · 2026-03-31
>
> Dear Reviewer dEMn,
>
> Thank you for your feedback! We address your concerns below:
>
> > The results echo previous work (e.g. Donahue et al 2022)...In contrast to these previous work, this paper introduces uncertainty, but it is not surprising that adding uncertainty makes complementarity more difficult
>
> A key message from our paper is that uncertainty makes a *qualitative* difference: Without uncertainty, complementarity only requires that the AI signal contain *any* information not in the human signal. With uncertainty, the AI signal must “cancel out” human errors, which is a much more stringent condition. Our empirical results show that current LLMs satisfy the first condition but fail the second. Donahue et al (2022) essentially provides a description within the no-uncertainty case of when potentially non-optimal ways of combining human and AI signals may still achieve complementarity. We have updated the Related Work section to more clearly articulate this core distinction.
>
> > The core results use a Gaussian model…I found the non-Gaussian setting to be much more compelling and wish the paper expanded on this
>
> Thank you for this suggestion! We would like to clarify that Proposition 3.9 and Proposition 3.10 establishes that the core results hold in a much more general non-Gaussian setting; our initial focus on the Gaussian setting was largely to build intuition. Following your suggestion, we have updated the draft to show further results on the non-Gaussian setting by default: We now include synthetic experiments for a non-Gaussian setting (https://anonymous.4open.science/r/y-1/fig.pdf). Importantly, we find the same takeaways hold: with negative error dependence, the robust estimator more strongly outperforms the expert-only baseline, while with positive error dependence, the robust estimator matches the expert-only baseline. See below for experimental setup details:
>
> **Details on numerical setting**: We use non-linear signals $\theta\sim t_5$ from a Student’s $t$ distribution.  We then define monotone nonlinear human and AI signals
> $$\phi_H=f(\theta)+\varepsilon_H,\ \phi_{AI}=g(\theta)+\varepsilon_{AI},$$
> where
> $$f(\theta)=\theta+0.5\tanh(\theta),\ g(\theta)=0.5\theta+0.5\tanh(\theta).$$
> For the noise terms, we draw $\varepsilon_H\sim\mathrm{Laplace}(0,0.75)$, $\eta\sim\mathrm{Laplace}(0,0.75)$, and set
> $$\varepsilon_{AI}=\alpha\tanh(\varepsilon_H)+\eta.$$
> Under this construction,
> $E[\varepsilon_{AI}\mid \varepsilon_H = e] = \alpha \tanh(e)$
> so $\alpha<0$ gives negative dependence and $\alpha>0$ positive. We generate \(n=10,000\) samples, split into 10% calibration and 90% evaluation sets. On the calibration split, we estimate the human-only predictor and the residualized AI signal. For our non-parametric estimators, we use Gaussian-kernel Nadaraya-Watson regression, with a bandwidth selected using Silverman’s rule of thumb.
>
> > How might results depend on estimation error? If obtaining realizations of the state is expensive, the human might also have uncertainty about the quality of their own predictions; would be interesting to see if/how the problem changes in this case.
>
> This is an interesting point! We have updated the draft to add this discussion. The most direct route for handling estimation error within our framework is to model the human as having a prior over their own prediction quality and performing a Bayesian update given samples of the true state. From a decision making perspective, all that matters at that point is the human’s predictive posterior $p(\theta|\phi_H)$, which accounts for any intermediate uncertainty in the quality of $\phi_H$. As long as that posterior update lies in one of the distributional classes our analysis covers, nothing changes from a formal perspective. A similar argument applies to approximation of the joint distribution over $(\phi_H, \phi_{AI})$.
>
> In the non-Gaussian case, the analysis is entirely unchanged for almost any set of Bayesian updates over signal quality: the induced $p(\theta, \phi_H)$ should simply satisfy the same assumptions as before (finite second moments for the analysis to apply, negatively dependent errors with the AI in the sense of Equation 8 to guarantee complementarity). By the CLT, it is natural to model the update as Gaussian distributed (see e.g. Che and Namkoong 2023 https://arxiv.org/abs/2303.11582), so in the Gaussian case a conjugate prior would leave the analysis unchanged as well.
>
> We thank the reviewer again for their time and consideration. We are more than happy to address any other concerns they might have.

---

> > ### Author Rebuttal · Reviewer_dEMn · 2026-04-03
> >
> > Thank you for the response and clarifications! I think the changes will improve the paper, and I have updated my score.

---

### Official Review · Reviewer_vwmu · 2026-03-13

**Soundness:** 4
**Presentation:** 3
**Significance:** 4
**Originality:** 3
**Overall Recommendation:** 5
**Confidence:** 3

**Summary:**

This paper contributes to the theory in human-AI collaboration, answering the questions for what are the necessary properties for a robust complementary decision rule to exist. The paper builds a series of propositions to characterize such properties, and therefore produces a robust estimator to beat the human-alone decisions under all possible distributions. The authors validate the effectiveness of the robust estimator on synthetic datasets. The real-world experiments find that the real AIs and humans fail to fulfill the conditions for robust complementarity, highlighting future works for targeting the design and evaluation of AI at complementary.

**Compliance With Llm Reviewing Policy:**

Affirmed.

**Key Questions For Authors:**

- What if the state does not follow Gaussian distribution? Would that change any theoretical results in the paper? Moreover, the real-world datasets do not necessary have Gaussian state. Would it be a confounder of the findings (such as would it make the method wrongly judge the chance to achieve robust complementary)?
- In assumption 3.1, what is the reason that the authors only consider the uncertainty in AI signals but not human signals? What would be an example of that case? How can the results be extended to the case that considers the uncertainty in human?

**Limitations:**

yes.

**Strengths And Weaknesses:**

Strengths:
- The problem of uncertainty in human-AI collaboration is well-motivated. Push forward the boundary of human-AI collaboration research.
- The theory is easy to follow, and offers impactful takeaways.
- I found the evaluation is comprehensive for a theory-prospective paper. The study on real-world datasets also offers very interesting findings.

Weaknesses:
- The setup is simplified a lot, where the signals from human and AI are predictions on the state. It may overlook some cases such as the predictions are not calibrated (human or AI have bias) or the state is not Gaussian. Are these valid concerns? If so, it may worth explicitly discussing such problems in the introduction.

Typos I found:
- "if it's prediction carries..." -> "if its prediction carries..." on line 34 on right column
- "They key feature of our..." -> "The key feature of our..." on line 114 on left column

---

> ### Author Rebuttal · Authors · 2026-03-31
>
> Dear Reviewer vwmu,
>
> Thank you for your feedback! We address your concerns below:
>
> > The setup is simplified a lot, where the signals from human and AI are predictions on the state.  It may overlook some cases such as the predictions are not calibrated or the state is not Gaussian. Are these valid concerns? If so, it may worth explicitly discussing such problems in the introduction.
>
> Thank you for this clarification question! We would like to note that in Section 3.2, our framework addresses a more general class of signals, which already encompasses non-Gaussian states. Specifically, Propositions 3.9 and 3.10 establish our main results without a Gaussian assumption on $\theta$.
>
> With respect to miscalibration, note that our model explicitly allows bias/miscalibration. We illustrate in the Gaussian setting for simplicity, since the non-Gaussian setting is strictly more expressive: Suppose that the machine signal has $\lambda_{AI} = c$. Then, $E[\theta|\phi_{AI}]] = \frac{1}{c} \phi_{AI}$, so the machine signal is miscalibrated (calibration would require that $E[\theta|\phi_{AI}]] =  \phi_{AI}$).
>
> > What if the state does not follow Gaussian distribution? Would that change any theoretical results in the paper? Moreover, the real-world datasets do not necessary have Gaussian state. Would it be a confounder of the findings (such as would it make the method wrongly judge the chance to achieve robust complementary)?
>
> We consider the non-Gaussian case in Section 3.2, where Proposition 3.10 shows that the core theoretical results (condition for existence of a robustly improving decision rule) extend without parametric assumptions on the data-generating distribution (instead, we give nonparametric definitions of core conditions like “negatively correlated errors”).
>
> > In assumption 3.1, what is the reason that the authors only consider the uncertainty in AI signals but not human signals? What would be an example of that case? How can the results be extended to the case that considers the uncertainty in human?
>
> Great question! The motivation for our framework is that a human user who is considering adopting AI as a decision aid has past experience with their own success at a task, but faces uncertainty about the quality of the AI signal. For example, a human forecaster or scientist knows their own past track record. Our framework can indeed be extended so that the uncertainty set also contains a range of possible values for the correlation of the human signal with the ground truth. This would propagate into more uncertainty in the correlation of the residual r with the ground truth and narrow the range for robust complementarity.
>
> > Typos
>
> Thank you for catching the typos! We have fixed them in the draft.
>
> We thank the reviewer again for their time and consideration. We are more than happy to address any other concerns they might have.

---

> > ### Author Rebuttal · Reviewer_vwmu · 2026-04-02
> >
> > Thank you for your rebuttal. It fully answers my questions.

---

### Decision · Program_Chairs · 2026-04-30

**Decision:**

Accept (regular)

**Comment:**

This paper studies when AI assistance can robustly improve human decision making when the human is uncertain about the quality of the AI signal. The authors develop a decision-theoretic framework that characterizes when a decision rule can guarantee improvement over relying on the human alone, and show that negatively correlated human and AI errors play a central role in enabling such robust complementarity under uncertainty. The paper includes theoretical results for both Gaussian and more general settings, along with synthetic experiments and empirical analyses on real-world forecasting benchmarks.

The reviewers were overall positive and viewed the problem as important and relevant. The main concerns centered on the simplicity of the initial modeling assumptions, the need to better position the paper relative to related work on complementarity by design, and the need for clearer discussion of the experiments, limitations, and practical implications. In the rebuttal and discussion, the authors addressed these points by clarifying the problem setting and scope, strengthening the discussion of the non-Gaussian/general formulation, adding experimental details and additional results, and improving the paper’s positioning relative to prior work. These responses have substantially addressed the concerns raised by reviewers.  Therefore, I recommend accepting the work.